# All-trans retinoic acid changes muscle fiber type via increasing GADD34 dependent on MAPK signal

Yuichiro Adachi[1],*, Masashi Masuda[1],*, Iori Sakakibara[2], Takayuki Uchida[2], Yuki Niida[1], Yuki Mori[1], Yuki Kamei[1], Yosuke Okumura[1], Hirokazu Ohminami[1], Kohta Ohnishi[1], Hisami Yamanaka-Okumura[1], Takeshi Nikawa[2], Yutaka Taketani[1]

**All-trans retinoic acid (ATRA) increases the sensitivity to unfolded protein response in differentiating leukemic blasts. The downstream transcriptional factor of PERK, a major arm of unfolded protein response, regulates muscle differentiation. However, the role of growth arrest and DNA damage-inducible protein 34 (GADD34), one of the downstream factors of PERK, and the effects of ATRA on GADD34 expression in muscle remain unclear. In this study, we identified ATRA increased the GADD34 expression independent of the PERK signal in the gastrocnemius muscle of mice. ATRA up-regulated GADD34 expression through the transcriptional activation of *GADD34* gene via inhibiting the interaction of homeobox Six1 and transcription co-repressor TLE3 with the MEF3-binding site on the *GADD34* gene promoter in skeletal muscle. ATRA also inhibited the interaction of TTP, which induces mRNA degradation, with AU-rich element on *GADD34* mRNA via p-38 MAPK, resulting in the instability of *GADD34* mRNA. Overexpressed GADD34 in C2C12 cells changes the type of myosin heavy chain in myotubes. These results suggest ATRA increases GADD34 expression via transcriptional and post-transcriptional regulation, which changes muscle fiber type.**

## Introduction

The ER is a membranous organelle that has a central role in protein biosynthesis. Various stresses cause the accumulation of unfolded proteins in the ER, which induces ER stress (Szegezdi et al, 2006; Walter & Ron, 2011). An excessive ER stress response can result in various diseases, such as diabetes, inflammation, and cardiovascular diseases, including vascular calcification (Kim et al, 2008; Masuda et al, 2013, 2015). To alleviate ER stress, the unfolded protein response (UPR) is initiated by the activation of three ER transmembrane sensors: PKR-like endoplasmic reticulum kinase (PERK), inositol-requiring enzyme 1 (IRE1), and activating transcription

factor 6 (ATF6). PERK activation leads to the phosphorylation of a subunit of eukaryotic initiation factor 2 (eIF2α), resulting in the up-regulation of activating transcription factor 4 (ATF4) and C/EBP-homologous protein (CHOP).

UPR in skeletal muscle regulates muscle stem cell homeostasis and myogenic differentiation. Especially, the PERK pathway is one of the key signals to promote the progression and commitment of satellite cells to the myogenic lineage (Rayavarapu et al, 2012; Gallot et al, 2019). Deletion of *PERK*, *ATF4*, or *CHOP* in satellite cells inhibits myofiber formation via the decreased expression of myoblast determination protein 1 (MyoD) and myogenin, essential transcription factors to correctly differentiate in skeletal muscle after injury (Alter & Bengal, 2011; Gallot et al, 2019; Ebert et al, 2020). We recently reported that UPR is an important regulator of injured skeletal muscle of mice with chronic kidney disease (CKD) (Niida et al, 2020). Furthermore, sine oculis homeobox homolog 1 (Six1), is a mandatory transcription factor with co-activator eyes absent (Eya) and dachshund (Dach) or co-repressor Groucho/transducing-like enhancer of split (TLE) for myogenic differentiation (Jennings & Horowicz, 2008; Sakakibara et al, 2016; Maire et al, 2020). However, little is known about the relationship between Six1 and PERK-related gene in muscle differentiation.

ATF4 is the strong transcriptional activator of growth arrest and DNA damage-inducible protein 34 (GADD34/Ppp1r15a), with binding to its consensus sequence on the *GADD34* gene promoter. However, recent investigations have demonstrated that ATF4-independent transcription factors, hairy and enhancer of split1 (HES1) and nuclear matrix protein 4 (NMP4), also bind to *GADD34* gene promoter (Young et al, 2016; Lee et al, 2018). Because GADD34 dephosphorylates eIF2α by interacting with the catalytic subunit of type 1 protein serine/threonine phosphatase (PP1) (Harding et al, 2000; Pakos-Zebrucka et al, 2016), ATF4-mediated induction of GADD34 functions as a negative feedback loop of UPR by eIF2α dephosphorylation, which is essential for cell survival. Although GADD34 plays various roles without UPR signaling in some cells, such as cytokine production in dendritic cells, inhibition of apoptosis in liver cancer cells, and the response to chronic oxidative stress in

[1]Department of Clinical Nutrition and Food Management, Institute of Biomedical Sciences, Tokushima University Graduate School, Tokushima, Japan   [2]Department of Nutritional Physiology, Institute of Biomedical Sciences, Tokushima University Graduate School, Tokushima, Japan

Correspondence: masuda.masashi@tokushima-u.ac.jp
*Yuichiro Adachi and Masashi Masuda contributed equally to this work.

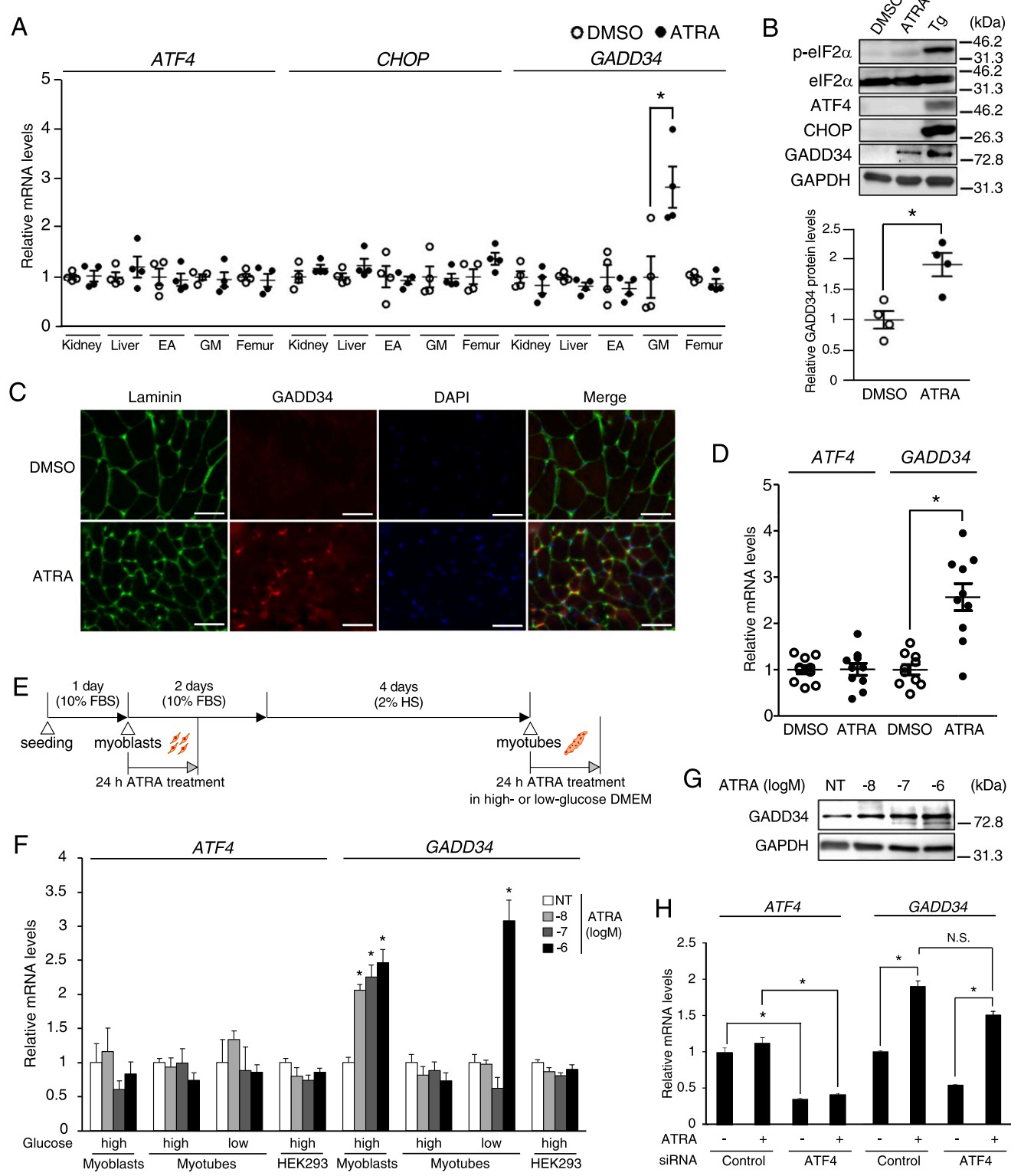

**Figure 1. Effects of all-trans retinoic acid (ATRA) treatment on PERK signaling in skeletal muscle.**
**(A, B, C)** 8-wk-old male C57BL/6J mice were given an intraperitoneal injection of 0.1% DMSO (vehicle) and ATRA (10 mg/kg) for 24 h. **(A)** The mRNA expression levels of *ATF4*, *CHOP*, and *GADD34* in kidney, liver, epididymal adipose, gastrocnemius muscle (GM), and femur of mice were determined by real-time PCR (*n* = 4 mice per group). **(B)** Western blotting of phosphorylated-eIF2α (p-eIF2α), total eIF2α (eIF2α), ATF4, CHOP, and GADD34 in GM (*n* = 4 mice per group). *$P < 0.05$ (two-tailed unpaired *t* test). **(C)** Immunofluorescence staining of Laminin (green), GADD34 (red), and DAPI (blue) in GM. Scale bars = 100 *μ*m. **(D)** The mRNA expression levels of *ATF4* and *GADD34* in primary myofibers isolated from 8-wk-old male C57BL/6J mice were incubated in the presence of 1 *μ*M ATRA or DMSO as a vehicle for 24 h (*n* = 10). *$P < 0.05$ (two-tailed

neurodegenerative diseases (Clavarino et al, 2012; Goh et al, 2018; Song et al, 2019), the specific role of GADD34 in skeletal muscle has not been revealed.

All-trans retinoic acid (ATRA), a metabolite of vitamin A, is a ligand for nuclear receptors, including retinoic acid receptors (RARα, β, and γ). RARs in the nucleus work as a transcription factor by forming a heterodimer with retinoid X receptors (RXRs) and binding to retinoic acid-response elements (RAREs) in target gene promoters (Mangelsdorf & Evans, 1995; Chambon, 1996). ATRA also activates various MAPK kinase cascades, such as p38 and ERK (p44/42), with RARα in membrane lipid rafts and RARγ in the cytosol (Eochette-Egly, 2015). In addition, activated p38 and ERK are involved in not only the transcriptional regulation of the target gene via translocation of the target gene to the nucleus but also the post-transcriptional regulation of the target mRNA by interacting with the mRNA-binding proteins (RBPs), such as tristetraprolin (TTP), which controls mRNA stability (Stoecklin et al, 2004; Bhattacharyya et al, 2011). ATRA has an important role in the differentiation and apoptosis of blood cells and the sensitivity to ER stress through the differentiation of human leukemic cells (Masciarelli et al, 2018). In contrast, ATRA supplementation ameliorates the ethanol-induced expression of ATF4 and CHOP in the rat liver (Nair et al, 2018). Interestingly, ATRA also controls muscle differentiation via the transcriptional regulation of MyoD and myogenin (Chen et al, 2015; Lamarche et al, 2015), whereas the effects of ATRA on PERK-related genes in skeletal muscle cells are still unknown.

In this study, we investigated the effects of ATRA on the PERK pathway in various organs, especially muscle. Our results demonstrate that ATRA increases the GADD34 gene expression via the down-regulation of Six1 and the stabilization of GADD34 mRNA in skeletal muscle. We also characterized the role of GADD34 in skeletal muscle fiber type change.

# Results

### ATRA induces GADD34 gene expression in the gastrocnemius muscle (GM) and C2C12 cells without translational activation of ATF4

To assess the response of the PERK signal to ATRA in various tissues, such as the kidney, liver, epididymal adipose (EA), GM, and femur, mice were treated with ATRA (total dose: 10 mg/kg body weight). The expression of ATF4 and CHOP mRNA were not changed in all tissues (as described above) of mice treated with ATRA for 24 h. ATRA treatment significantly increased the GADD34 mRNA expression only in the GM of mice (Fig 1A). Western blotting revealed that ATRA treatment did not increase phosphorylated eIF2α (p-eIF2α), ATF4, or CHOP downstream of the PERK signal in GM in comparison to vehicle; however, it increased GADD34 protein expression levels

(Fig 1B). Immunofluorescence staining of Laminin and GADD34 in GM showed that ATRA-induced GADD34 protein localizes in the area inside of Laminin-staining (myofiber) (Fig 1C). As a result of culturing primary myofibers from GM of C57BL/6J, ATRA did not change the ATF4 mRNA expression but increased the GADD34 mRNA expression same as the above in vivo test (Fig 1D). To determine whether the increase in the expression of GADD34 by ATRA is associated with the differentiation of skeletal muscle cells, C2C12 cells were gradually differentiated from myoblasts to myotubes, as shown in Fig 1E. Recent reports showed that there are cases in which low-glucose DMEM is better to get a similar with in vivo study than high-glucose DMEM (Luo et al, 2019; Abdelmoez et al, 2020). For this reason, C2C12 myotubes were cultured with ATRA in low- or high-glucose DMEM, respectively. In C2C12 myoblasts, myotubes, and HEK293 cells, the expression of ATF4 mRNA was comparable between DMSO and ATRA treatment, which was in line with the results of in vivo studies. However, ATRA increased the GADD34 mRNA and protein expression levels in C2C12 undifferentiated myoblasts or myotubes (low-glucose), but not in myotubes (high-glucose) or HEK293 cells (Fig 1F and G). We generated C2C12 cells with the knockdown of ATF4 to confirm the effects of ATRA on the GADD34 expression independently of ATF4; with the use of ATF4-specific siRNA, the endogenous ATF4 mRNA levels of these C2C12 cells were reduced by more than 50%. ATF4-knockdown did not inhibit the induction of the expression of GADD34 mRNA by ATRA (Fig 1H).

### ATRA up-regulates the transcriptional activity of the GADD34 gene in C2C12 cells

To investigate the molecular mechanisms underlying the undifferentiated muscle-specific regulation of the GADD34 gene expression by ATRA, we examined the responsiveness of human GADD34 gene promoters to ATRA using a luciferase assay with C2C12 cells and HEK293 cells. The overexpression of RARs increased the luciferase activity of pGADD34-0.5k in both C2C12 and HEK293 cells (Fig 2A). To explore the ligand-independent RAR activation of GADD34 gene promoter activity, several reporter constructs lacking portions of the 5'- and 3'-promoter regions of the human GADD34 genes were tested in C2C12 cells overexpressing RAR/RXR, with or without ATRA. These deletion analyses suggested that the human GADD34 gene has the RAREs (RARE1: −28 to −15 and RARE2: −15 to −1), but each RARE is an uncommon direct repeat (DR) with 2-bp spacer (DR2) or 3-bp spacer (DR3) (Fig S1A and B). Although the classical RARE is a DR with 5-bp (DR5), RAR/RXR heterodimers also bind to DR2 or DR3 (Cunningham & Duester, 2015). Mutation analyses suggested that the RARE2 is responsible for ATRA-independent RAR activation of the human GADD34 promoter activity (Fig S1C). ATRA increased the luciferase activity of pGADD34-0.5k in C2C12 cells expressing RARα and RARγ, but not RARβ (Fig 2A). Surprisingly, TTNPB, a major agonist of RAR, did not increase the GADD34 gene

---

unpaired t test). **(E)** Schematic illustration of the experimental timeline in C2C12. C2C12 cells were cultured with high-glucose DMEM except ATRA treatment situation. **(F)** The mRNA expression levels of ATF4 and GADD34 in C2C12 cells (myoblasts and myotubes) and HEK293 cells treated with the indicated concentrations of ATRA (n = 3–4). Myotubes were cultured with high- or low-glucose DMEM respectively under ATRA treatment. *P < 0.05 versus DMSO (NT) (one-way ANOVA with a Student–Newman post hoc test). **(G)** Western blotting of GADD34 in C2C12 cells myoblasts with the indicated concentrations of ATRA. **(H)** The mRNA expression levels of ATF4 and GADD34 in C2C12 myoblasts transfected with ATF4 siRNA (siATF4) or control and incubated in the presence of 1 μM ATRA or DMSO as a vehicle for 24 h (n = 3–4). Data are presented as the mean ± SEM. *P < 0.05 versus DMSO (NT) (one-way ANOVA with a Student–Newman post hoc test). N.S., not significant. Source data are available for this figure.

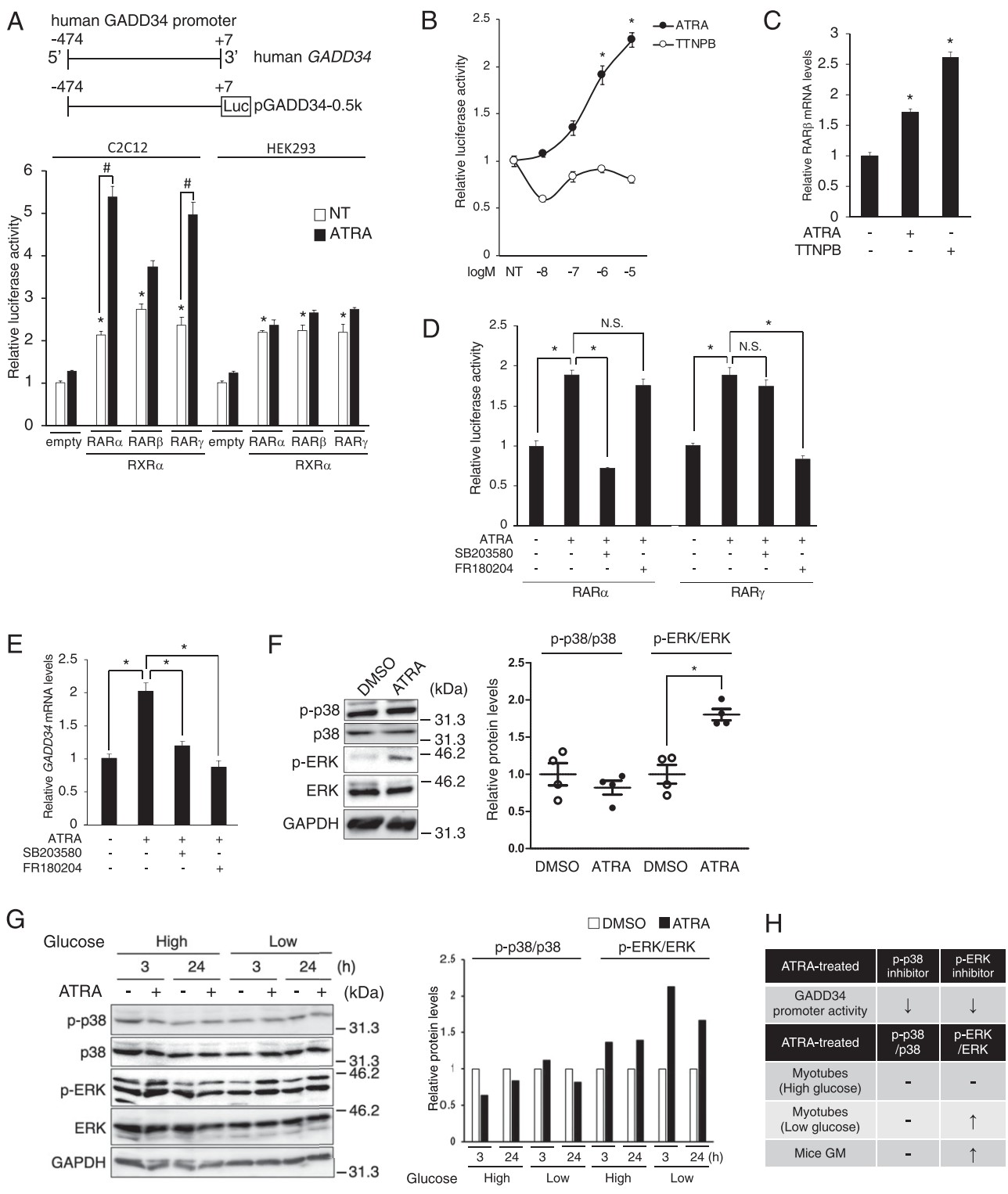

**Figure 2.   Activation of human *GADD34* gene promoter by all-trans retinoic acid (ATRA) and its receptors through MAPK signals in C2C12 cells.**
**(A)** A schematic illustration of the human *GADD34* gene promoter in the upper panels. pGADD34-0.5k and pCMV-β were transfected with pSG5-RAR (α, β, γ) and pSG5-RXRα, or empty vector and incubated in the presence of 1 μM ATRA or DMSO (NT) as a vehicle for 24 h in C2C12 and HEK293 cells (*n* = 3–4). *P < 0.05 versus empty vector. #P < 0.05 (one-way ANOVA with a Student–Newman post hoc test). **(B, D)** pGADD34-0.5k, pSG5-RXRα, pCMV-β, and (B) pSG5-RARα, (D) pSG5-RARα or pSG5-RARγ were transfected and incubated in the (B) indicated concentrations of ATRA (black circles) or TTNPB (white circle) or (D) in the presence of 1 μM ATRA or DMSO (NT) as a vehicle and 10 μM SB203580 (p38 MAPK inhibitor) or 10 μM FR180204 (ERK inhibitor) for 24 h in C2C12 myoblasts. *P < 0.05 versus NT (one-way ANOVA with a Student–Newman post hoc test)

promoter activity, although TTNPB increased the mRNA expression of the target gene (Fig 2B and C) (Thé et al, 1990). Unlike TTNPB, ATRA can activate the non-genomic p38 and ERK MAPK signals via extranuclear RAR$\alpha$/$\gamma$ and RAR$\gamma$, respectively (Bouchard & Paquin, 2013; Tanoury et al, 2013; Khatib et al, 2019). Next, we tested whether these MAPK signals are involved in the regulation of the GADD34 expression by ATRA. As a result, the ATRA-induced GADD34 gene promoter activity and the expression of GADD34 mRNA were blocked by p38 inhibitor SB203580 or ERK inhibitor FR180204 via RAR$\alpha$/$\gamma$ or RAR$\gamma$, respectively (Fig 2D and E). As shown in Fig 1F, ATRA increased GADD34 mRNA expression in C2C12 myotubes cultured with low-glucose DMEM but not high-glucose. These results suggested that ATRA-induced MAPK activation leads to those differences in the sensitivity to ATRA. As expected, ATRA induced the phosphorylation of ERK, unlike p-38, in GM and C2C12 myotubes cultured with low-glucose DMEM, but not high-glucose (Fig 2F and G). These results suggest that ATRA increases GADD34 gene promoter activity via ERK phosphorylation in skeletal muscle (Fig 2H).

### Six1 down-regulates the transcriptional activity of the GADD34 gene via MEF3-binding site in C2C12 cells

To explore the muscle-specific regulatory element of ATRA on the human GADD34 gene promoter, several reporter constructs lacking portions of the 5′-promoter region of the human GADD34 genes were tested in C2C12 cells overexpressing RAR/RXR, with or without ATRA. These deletion analyses suggest that the sequence from −162 to −131 is responsible for ATRA-dependent activation of the human GADD34 gene promoter activity in C2C12 cells (Fig 3A). Next, we searched for undifferentiated muscle-specific transcription factor binding sites from −162 to −131 on the human GADD34 gene promoter with MatInspector-Genomatix. A search for transcription factor binding motifs with this region suggested five potential binding sites: three MEF3 sites, Pax3, and Sox6 (Fig 3B). Based on this search, we overexpressed Six1, a major transcription factor for the MEF3-binding site, Pax3, and Sox6 with GADD34 gene promoter constructs in C2C12 cells. Although each transcription factor decreased the luciferase activity of pGADD34-0.16k, Six1 did not affect the activity of pGADD34-0.13k, unlike Pax3 and Sox6 (Fig 3C). The overexpression of Six1 reduced the GADD34 mRNA and protein expression levels in C2C12 cells (Fig 3D and E). To investigate which MEF3-binding sites are responsible for the decrease in GADD34 gene promoter activity induced by Six1, we made luciferase reporter vectors of three mutated MEF3-binding sites in the pGADD34-0.13k, as shown in Fig 3F. These mutation analyses showed that the MEF3-2 sequence in the MEF3-binding sites is essential for the repression of the GADD34 gene promoter activity by Six1 (Fig 3G). Next, we examined whether Six1 binds to the MEF3-2 sequence on the GADD34 gene promoter with an EMSA analysis. A radiolabeled

oligonucleotide containing a GADD34-MEF3 probe, but not a mutated GADD34-MEF3 probe (GADD34-Mut-M2), detected a band in nuclear extracts prepared from C2C12 cells overexpressing Six1. Although these complexes were susceptible to competition with unlabeled GADD34-MEF3 and MEF3-2 (MB), an unlabeled GADD34-Mut-M2 (M2), MEF3-1 (MA), and MEF3-3 (MC) did not compete with these complexes (Fig 3H and I). These results suggest that Six1 may be involved in the ATRA-induced GADD34 expression through the MEF3-binding site in the GADD34 gene promoter.

### ATRA up-regulates the human GADD34 gene expression through the reduction of the Six1 expression in C2C12 cells

Next, we investigated the effects of ATRA on the Six1 expression in several tissues of mice. The Six1 mRNA levels were undetectable in the kidney, liver, and EA. ATRA treatment suppressed the mRNA expression of Six1 (which is highly expressed only in the GM of mice, as well as the protein expression of Six1 [Fig 4A and B]). The mRNA expression levels of Pax3 and Sox6, which affect GADD34 gene promoter activity, were not changed in the GM of ATRA-treated mice in comparison to mice treated with DMSO (Fig S2A). Likewise, ATRA treatment down-regulated the of Six1 mRNA and protein expression in C2C12 myoblasts or myotubes (Fig 4C and D). We also examined the time-dependent effects of ATRA on the mRNA expression of Six1 and GADD34 for up to 24 h in C2C12 cells. In comparison to DMSO, ATRA transiently decreased the expression of Six1 mRNA only after 24 h of treatment. It also increased the expression of GADD34 mRNA; this was observed at both 3 and 24 h (Fig 4E). Moreover, Six1-knockdown increased the GADD34 protein expression and inhibited ATRA-induced GADD34 protein expression in C2C12 cells (Fig 4F). The mutated promoter constructs shown in Fig 3F were transfected in C2C12 cells to examine the effects of ATRA and Six1 on MEF3-binding sites on the GADD34 gene promoter. These mutation analyses suggested that the binding of Six1 to the MEF3-2 sequence is essential for the activation of the GADD34 gene promoter by ATRA (Figs 4G and S2B). An EMSA assay showed that the band was detected by a contact with a radiolabeled oligonucleotide containing a GADD34-MEF3 probe and nuclear extracts prepared from C2C12 cells treated with ATRA or Six1-knockdown was decreased in comparison to treatment with DMSO or control (Fig 4H). We also investigated whether the Six1 binds to MEF3 sequence on mouse GADD34 promoter by ChIP assay with four primer pairs. As a result, Six1 binds to mouse GADD34 promoter detected by primer pair No. 2 and 4, which indicates Six1 binds to GADD34 promoter beyond the species (Fig S2C and D). As shown in Fig 2H, because ERK signal is involved in the increase of the GADD34 expression by ATRA, we tested whether these MAPK signals regulate the mRNA expression of Six1 in C2C12 cells. As a result, ERK inhibitor blocked the decrease of the Six1 mRNA expression induced by ATRA (Fig 4I). These results indicated

---

Each point represents the average of quadruplicate analysis ± SEM normalized for $\beta$-gal activity ($n$ = 3–4). N.S., not significant. **(C, E)** The mRNA expression levels of (C) *RAR$\beta$*, (E) *GADD34* in C2C12 myoblasts treated with 1 $\mu$M ATRA, (C) 1 $\mu$M TTNPB, (E) 10 $\mu$M SB203580, 10 $\mu$M FR180204, or DMSO (NT) as a vehicle. *$P$ < 0.05 versus NT (one-way ANOVA with a Student–Newman post hoc test). **(F, G)** Western blotting of phosphorylated p38 (p-p38), total p38 (p38), phosphorylated ERK (p-ERK), and total ERK (ERK) in (F) gastrocnemius muscle ($n$ = 4 mice per group), (G) C2C12 myotubes cultured in high- or low-glucose DMEM each incubated in the presence of 1 $\mu$M ATRA or DMSO as a vehicle for 3 or 24 h. Data are presented as the mean ± SEM. *$P$ < 0.05 (two-tailed unpaired $t$ test). **(H)** Table with the following information of ATRA-induced p38 and ERK phosphorylation in each cells and mice. Source data are available for this figure.

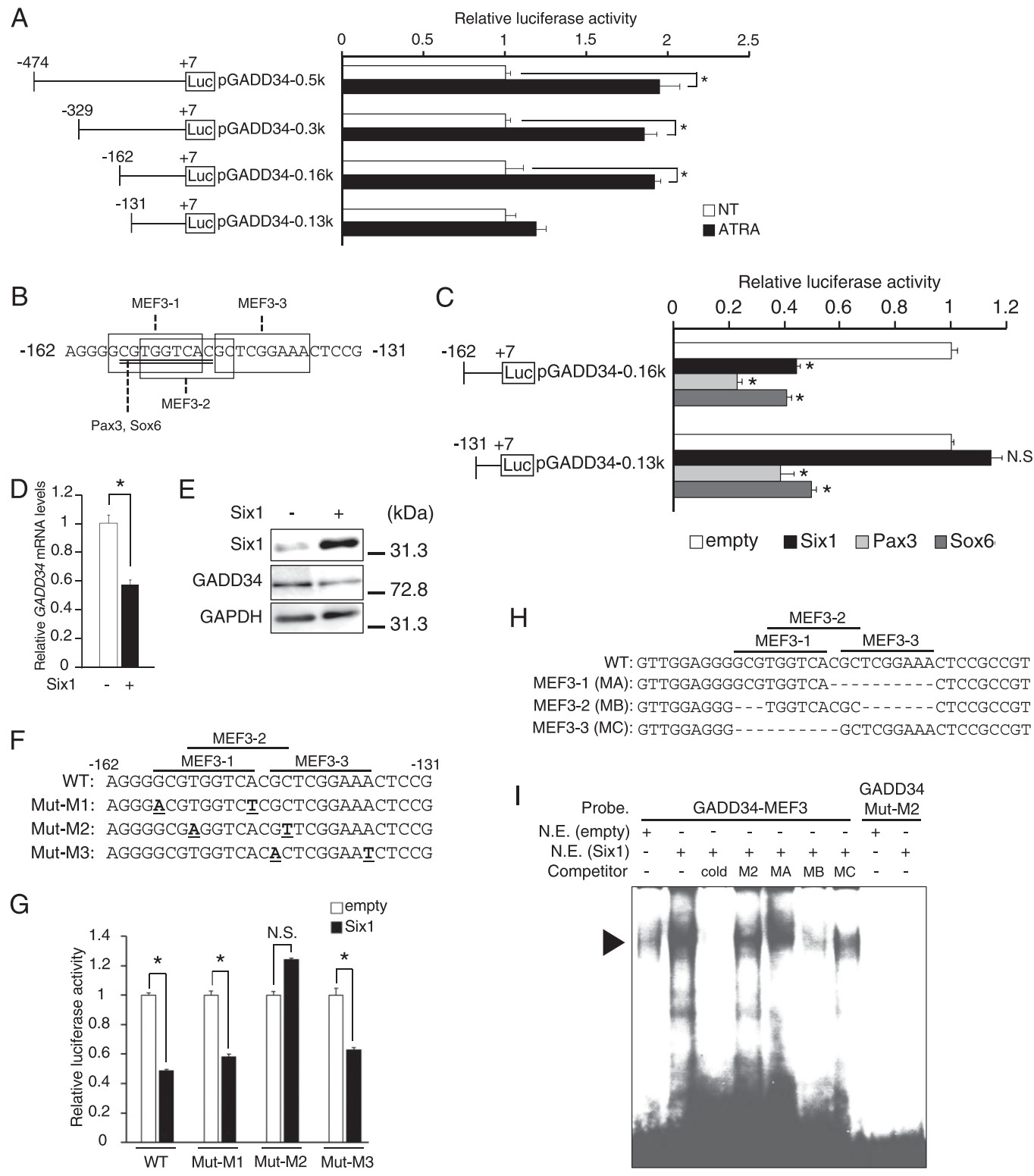

**Figure 3.  Effects of Six1 on the MEF3 site and the mutation analysis of the *GADD34* gene promoter.**
**(A)** Transcriptional activity of deletion constructs of *GADD34* gene promoters (pGADD34-0.5k, pGADD34-0.3k, pGADD34-0.16k, and pGADD34-0.13k). Deletion constructs are illustrated in the panels on the left. C2C12 cells were transfected with the indicated *GADD34* gene reporter constructs, pSG5-RARα and pSG5-RXRα, and incubated in the presence of 1 μM all-trans retinoic acid or DMSO (NT) as a vehicle for 24 h (*n* = 3–4). *P < 0.05 (two-tailed unpaired *t* test). **(B)** Myoblast-specific transcription factor binding sites (Pax3 and Sox6 sequences: underline, MEF3 sequences: box) in the human *GADD34* gene promoter region. **(C)** Each *GADD34* reporter plasmid (pGADD34-0.16k and pGADD34-0.13k) was transfected with each pCR3 plasmid (empty, Six1, Pax3, or Sox6) into C2C12 myoblasts (*n* = 3–4). *P < 0.05 versus empty vector (one-way

that ATRA suppresses the binding of Six1 with the MEF3 sequence on the *GADD34* gene promoter by decreasing the expression of Six1, through the ERK signal, resulting in the increased GADD34 expression in skeletal muscle.

### ATRA up-regulates the transcriptional activity of the *GADD34* gene via co-repressor TLE3 with Six1 in C2C12 cells

Six1 requires the Eya family as the co-activator for transcriptional activation but requires the TLE family as the co-repressor for transcriptional repression (Jennings & Horowicz, 2008). To define a co-repressor that is responsible for the Six1-mediated reduction of the *GADD34* gene expression, we first analyzed the mRNA expression levels of the TLE family (*TLE1-6*) by real-time qPCR with the absolute standard curve method. Among the TLE family, the *TLE1*, *TLE3*, and *TLE5* genes were expressed in C2C12 cells (Fig 5A), we generated C2C12 cells with the knockdown of these genes using specific siRNA for each of these genes. As shown in Fig 5B, *TLE1*, *TLE3*, and *TLE5* siRNAs reduced the expression of these genes by 50%, 80%, and 99%, respectively. Although pGADD34-0.16k significantly increased the luciferase activity in response to *TLE3*-knockdown—but not *TLE1*- and *TLE5*-knockdown—pGADD34-0.13k exhibited no increase in response to these knockdowns (Fig 5C). The mutated promoter constructs shown in Fig 3F revealed that the MEF3-2 sequence is essential for the activation of the *GADD34* gene promoter by *TLE3*-knockdown (Fig 5D). Overexpression of Six1, *Six1*-Knockdown, or ATRA treatment does not change the luciferase activity of pGADD34-0.16k under *TLE3*-knockdown situations (Fig 5E). On the other hand, ATRA did not change the mRNA expression of *TLE1*, *TLE3*, and *TLE5* in C2C12 cells (Fig 5F). These results indicate that TLE3 works as a co-repressor with Six1 for the *GADD34* gene expression induced by ATRA.

### ATRA increases the stability of *GADD34* mRNA via TTP in a p38-dependent manner

Thus far, we have demonstrated that ATRA increases the GADD34 expression by transcriptional regulation through the reduced expression of Six1. However, in comparison to vehicle, ATRA transiently increased the *GADD34* mRNA expression after 3 h of treatment, unlike the *Six1* mRNA expression (Fig 4E). Similarly to the mRNA expression, Western blotting revealed that the GADD34 protein expression was increased by ATRA treatment at 3 h in C2C12 cells (Fig 6A). These results imply the presence of the regulation of GADD34 expression by ATRA at an early time point, independently of the action of Six1. As expected, ATRA promoted *GADD34* mRNA stability at 3 h in C2C12 cells that were treated with actinomycin D, a transcriptional inhibitor (Fig 6B). Generally, mRNA stability is regulated through an AU-rich element (ARE: AUUUA motif) in the 3′-

untranslated region (3′UTR) of mRNA. To investigate the molecular mechanism underlying the *GADD34* mRNA stabilization in response to ATRA via the 3′UTR of its mRNA, we created a pGL3-GADD34-3′UTR construct (GADD34-3′UTR) by replacing the 3′UTR of the pGL3-basic with the 3′UTR of the *GADD34* gene, which contained two AREs, as illustrated in Fig 6C. ATRA dose-dependently stimulated the luciferase activity of the GADD34-3′UTR, but not pGL3-basic empty (control), in C2C12 cells (Fig 6C).

Because the mRNA stability requires signal transduction pathways of MAPK, such as p38 and ERK in lung cancer cells (Bhattacharyya et al, 2011), we examined which MAPK pathway is responsible for the stabilization of *GADD34* mRNA by ATRA using p38 and ERK inhibitors. Unlike ERK1/2 inhibitor, p38 inhibitor completely suppressed the ATRA-induced luciferase activity of pGADD34-3′UTR (Figs 6D and S3A). The phosphorylation of p38 inhibits the TTP contact with ARE, resulting in the induction of target mRNA stabilization (Bhattacharyya et al, 2011). Although ATRA and TNFα, a p38 phosphorylation inducer, stimulated the luciferase activity of GADD34-3′UTR, the effects of these treatments on its activity in C2C12 cells were canceled by *TTP*-knockdown (Figs 6E and S3B–D). In addition to TTP, human antigen R (HuR), one of the major RBPs, induces mRNA degradation (Stoecklin et al, 2004). Unlike *TTP*-knockdown, *HuR*-knockdown did not change the luciferase activity of GADD34-3UTR induced by ATRA (Figs 6E and S3E). To further test whether the ARE1 and ARE2 sequences are responsible for the regulation of *GADD34* mRNA stability by ATRA, we determined the luciferase activity of the mutated ARE1 (Mut-A1) and mutated ARE2 (Mut-A2) in the GADD34-3′UTR (Fig 6F). These mutation analyses showed that both of AREs are important for the enhancement of the stabilization of the *GADD34* mRNA by ATRA (Fig 6F). Using an RNA ChIP assay, we confirmed that p38 inhibitor treatment recovered the ATRA-decreased binding between *GADD34* mRNA and TTP (Fig 6G). These results suggest that ATRA suppresses TTP-induced *GADD34* mRNA degradation early time through the binding of ARE on the 3′UTR of the *GADD34* mRNA in a p38-dependent manner.

### GADD34 decreases the expression of MYHC2a and changes the muscle fiber type

Finally, C2C12 cells transfected with human GADD34 expression vector or empty (control) were collected on days 0, 1, 2, and 4 to examine the effects of GADD34 on the C2C12 phenotype (Figs 7A and S4B). Unexpectedly, the overexpression of GADD34 did not affect the thickness of myotubes, differentiation speed, mRNA expression of major muscle differentiation marker genes (*Pax7*, *Myf5*, *MyoD*, and *Myogenin*) or muscle-specific E3 ubiquitin ligase (*MuRF1* and *Atrogin1*), or the synthesis of proteins in C2C12 cells (Figs 7B–D and S4C and D). Although the GADD34 overexpression did not change

---

ANOVA with a Student–Newman post hoc test). **(D)** The mRNA expression levels of *GADD34* in C2C12 myoblasts transfected with each pCR3 plasmid (empty or Six1) (*n* = 3–4). **(E)** Western blotting of Six1 and GADD34 in C2C12 myoblasts transfected with each pCR3 plasmid (empty or Six1). **(F)** Three MEF3 sites mutated in the *GADD34* gene reporter region are underlined. Mut-M1, Mut-M2, and Mut-M3 targeted MEF3-1, MEF3-2, and MEF3-3, respectively. **(G)** Each *GADD34* reporter plasmid (WT, Mut-M1, Mut-M2, and Mut-M3) was transfected with pcDNA3 empty or Six1 into C2C12 myoblasts (*n* = 3–4). Data are presented as the mean ± SEM. *$P < 0.05$ (two-tailed unpaired *t* test). N.S., not significant. **(H)** Three oligonucleotides which contain specific MEF3 sequences (MEF3-1; MA, MEF3-2; MB, and MEF3-3; MC), respectively. **(I)** EMSAs using [32]P-labeled GADD34-MEF3 and mutated GADD34-MEF3-2 (GADD34-Mut-M2) as probes. EMSAs were performed with nuclear extracts (N.E.) from C2C12 myoblasts overexpressing Six1 or empty with the addition of unlabeled competitor oligonucleotides, as indicated. A 100-fold molar excess of each competitor was used. The location of the DNA-protein complex band is indicated by an arrowhead. cold, WT GADD34-Six1; M2, GADD34-Mut-M2; MA, MB, and MC are indicated in Fig 3F. Source data are available for this figure.

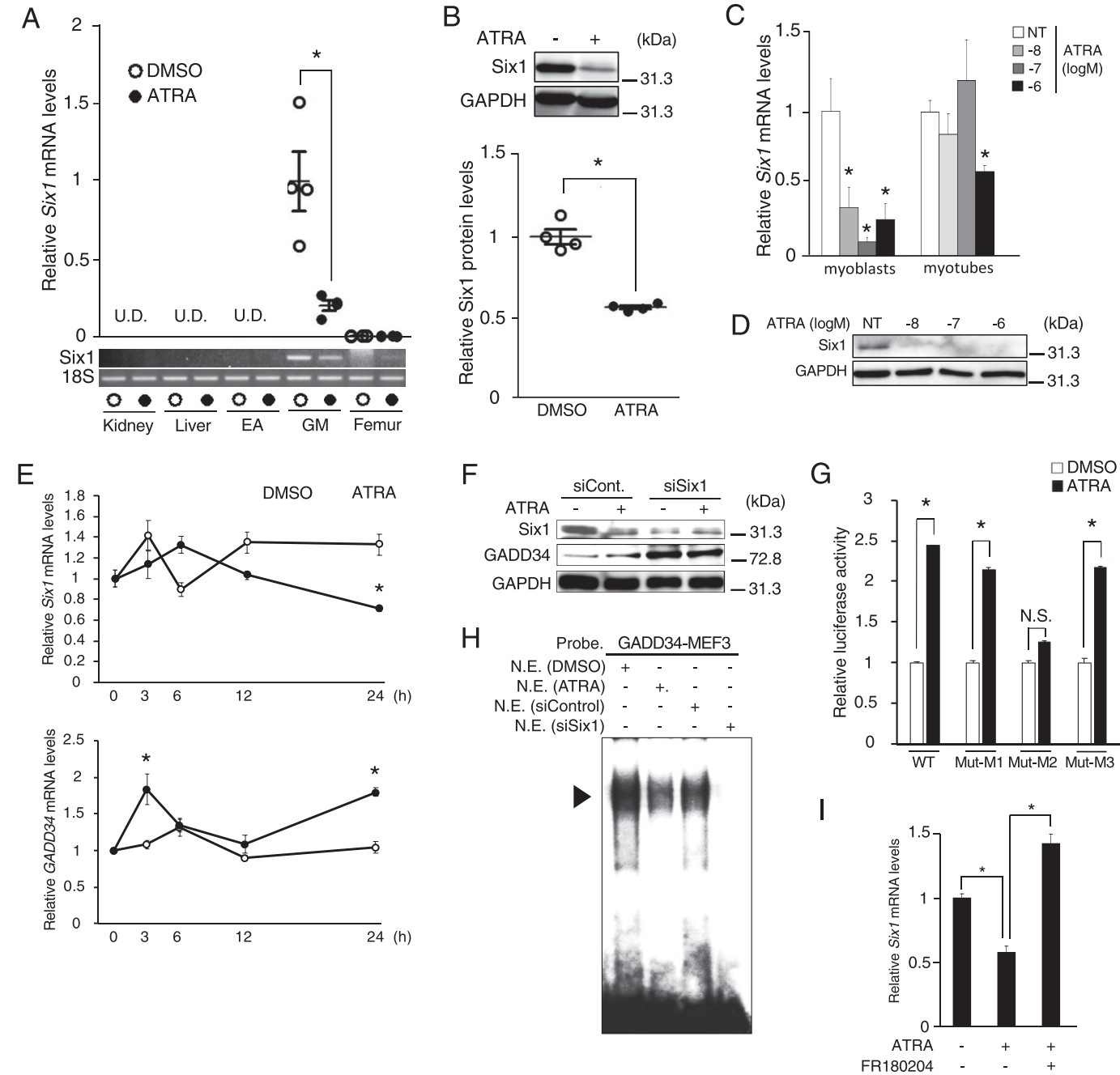

**Figure 4. Effects of all-trans retinoic acid (ATRA) treatment on Six1 and GADD34 expression in skeletal muscle.**
**(A)** The mRNA expression levels of *Six1* in kidney, liver, epididymal adipose, gastrocnemius muscle, and femur of ATRA-treated mice (*n* = 4 mice per group). *P < 0.05 (two-tailed unpaired *t* test). **(B)** Western blotting of Six1 in gastrocnemius muscle of ATRA-treated mice (*n* = 4 mice per group). *P < 0.05 (two-tailed unpaired *t* test). **(C)** The mRNA expression levels of *Six1* in C2C12 cells (myoblasts and myotubes) treated with the indicated concentrations of ATRA (*n* = 3–4). *P < 0.05 versus NT (DMSO) (one-way ANOVA with a Student–Newman post hoc test). **(D)** Western blotting of Six1 in C2C12 myoblasts with the indicated concentrations of ATRA. **(E)** The mRNA expression levels of *GADD34* and *Six1* in each C2C12 cells (0, 3, 6, 12, and 24 h) incubated in the presence of 1 µM ATRA or DMSO as a vehicle (*n* = 3–4). *P < 0.05 (two-tailed unpaired *t* test). **(F)** Western blotting of Six1 and GADD34 in C2C12 myoblasts transfected with *Six1* siRNA (siSix1) or control (siCont.) and incubated in the presence of 1 µM ATRA or DMSO as a vehicle for 24 h. **(G)** Each *GADD34* reporter plasmid (WT, Mut-M1, Mut-M2, and Mut-M3) was transfected and incubated in the presence of 1 µM ATRA or DMSO (NT) as a vehicle for 24 h in C2C12 myoblasts (*n* = 3–4). *P < 0.05 (two-tailed unpaired *t* test). **(H)** EMSAs using [32]P-labeled GADD34-MEF3 as probes. EMSAs were performed with nuclear extracts (N.E.) from C2C12 myoblasts treated with 1 µM ATRA or DMSO as a vehicle for 24 h, and transfected with *Six1* siRNA (siSix1) or control (siCont.). The location of the DNA-protein complex band is indicated by an arrowhead. **(I)** The mRNA expression levels of *Six1* in C2C12 myoblasts treated with 1 µM ATRA, 10 µM FR180204 or DMSO (NT) as a vehicle for 24 h. Data are presented as the mean ± SEM. *P < 0.05 (one-way ANOVA with a Student–Newman post hoc test). U.D., undetectable; N.S., not significant. Similar results were obtained from independent experiments. Source data are available for this figure.

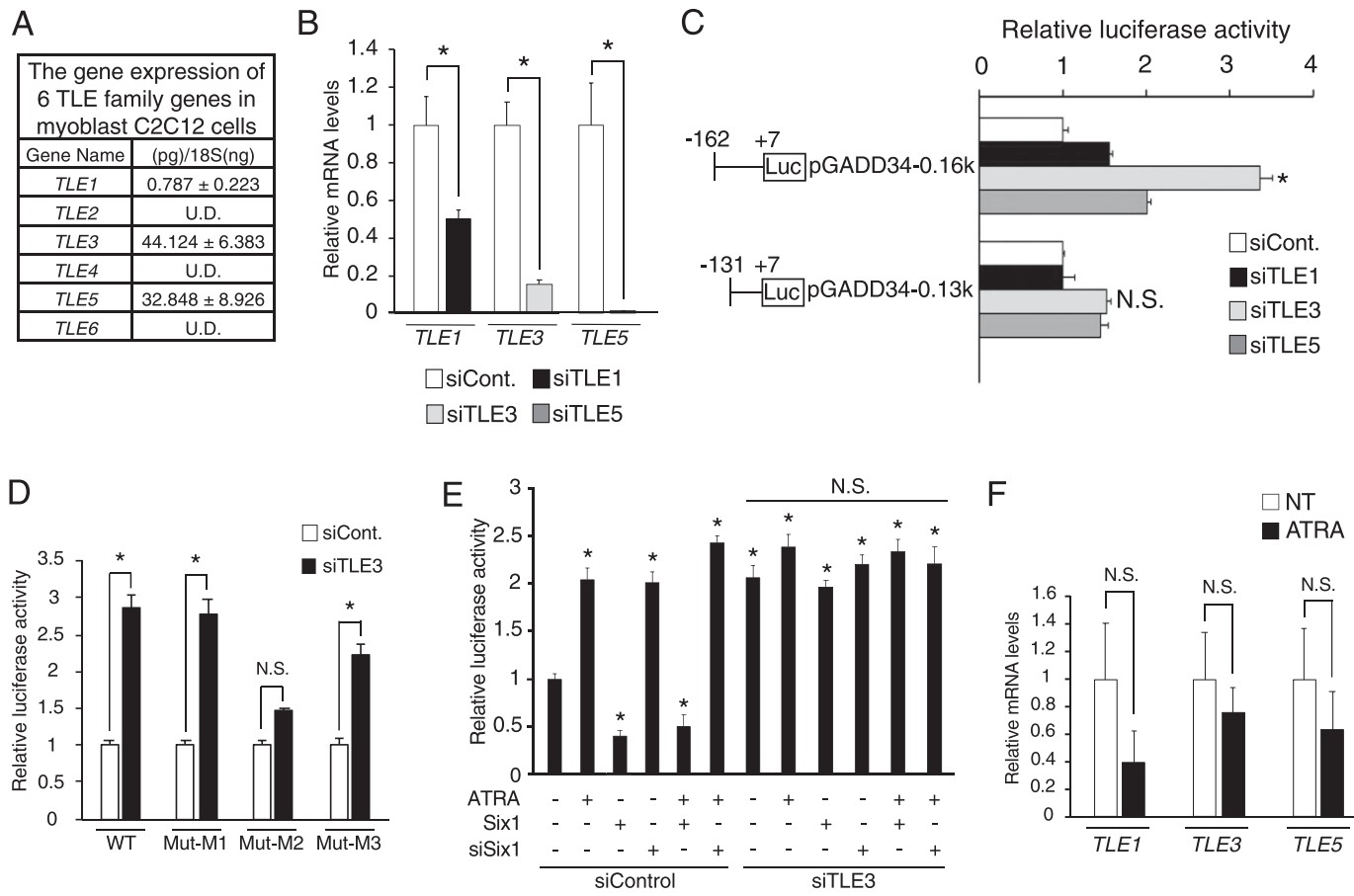

**Figure 5.    Effects of the TLE family on the *GADD34* gene promoter activity in C2C12 myoblasts.**
**(A)** The quantitative expression values of the TLE family (*TLE1–6*) were calculated using the absolute standard curve method of real-time PCR with plasmid templates containing each target gene (*n* = 3). **(B)** The mRNA expression levels of *TLE1, TLE3,* and *TLE5* in C2C12 myoblasts transfected with each siRNA (siCont., siTLE1, siTLE3, or siTLE5) (*n* = 3–4). *$P < 0.05$ (two-tailed unpaired *t* test). **(C)** Each *GADD34* reporter plasmid (pGADD34-0.16k and pGADD34-0.13k) was transfected with each siRNA (siCont, siTLE1, siTLE3, or siTLE5) into C2C12 myoblasts (*n* = 3–4). *$P < 0.05$ versus siCont. (one-way ANOVA with a Student–Newman post hoc test). **(D)** Each *GADD34* reporter plasmid (WT, Mut-M1, Mut-M2, and Mut-M3) was transfected with *TLE3* siRNA (siTLE3) or control into C2C12 myoblasts (*n* = 3–4). *$P < 0.05$ (two-tailed unpaired *t* test). **(E)** phGADD34-0.13k was transfected with each pCR3 plasmid (empty or Six1), *TLE3* siRNA (siTLE3), *Six1* siRNA (siSix1), or control and incubated in the presence of 1 *μ*M all-trans retinoic acid or DMSO as a vehicle for 24 h in C2C12 cells (*n* = 3–4). *$P < 0.05$ versus NT (one-way ANOVA with a Student–Newman post hoc test). **(F)** The mRNA expression levels of *TLE1, TLE3* and *TLE5* in C2C12 myoblasts treated with 1 *μ*M all-trans retinoic acid (two-tailed unpaired *t* test). Data are presented as the mean ± SEM. U.D., undetectable; N.S., not significant.

the total MYHC protein expression in C2C12 cells from Day 0 to 4, the protein expression of MYHC1 (type 1 slow fibers) was slightly increased in C2C12 cells that overexpressed GADD34 on Day 4. In contrast, the overexpressed GADD34 slightly decreased the protein expression of MYHC2 (type 2 fast fibers) and MYHC2 isoforms, such as MYHC2a and MYHC2x, but not MYHC2b, in C2C12 cells on Day 4 (Fig 7E). In addition, the overexpression of GADD34 down-regulated the mRNA expression of *MYH2* and *MYH1*, which respectively encode MYHC2a and MYHC2x proteins (Fig 7F). Moreover, ATRA treatment decreased the protein expressions of total MYHC, MYHC2, MYHC2a, and MYHC2x in the GM of C57BL/6J mice (Fig 7G).

## Discussion

In the present study, we showed the molecular mechanisms by which ATRA increases the GADD34 expression independently of ATF4 (known to be a potent inducer of GADD34) in skeletal muscle.

Likewise, although a recent study reported that GADD34 levels were decreased in satellite cells collected from injured muscle with increased ATF4 (Xiong et al, 2017), the mechanism of this paradoxical result is unknown. Because muscle atrophy can occur with differential sensitivity due to selective skeletal muscle fiber subtypes in various pathological conditions, muscle fiber type change is suggested to be an important factor for muscle wasting (Wanga & Pessinn, 2013). Muscle atrophy is a frequent complication in CKD patients (Carrero et al, 2013). Interestingly, the plasma levels of vitamin A (retinol or ATRA) increase in CKD patients (Gueguen et al, 2005; Jing et al, 2016). It has been reported that the percentage of type 1 slow fiber in type 2 fast fiber increases in CKD mice through a decrease in the expression of MYHC2a (Tamaki et al, 2014). However, the molecular mechanisms underlying the muscle fiber type change in CKD are largely unknown. In the present study, we revealed that the overexpression of GADD34 increases the MYHC1 protein expression, which is expressed in type 1 slow fibers, and decreases the MYHC2 protein expression, which is expressed in type

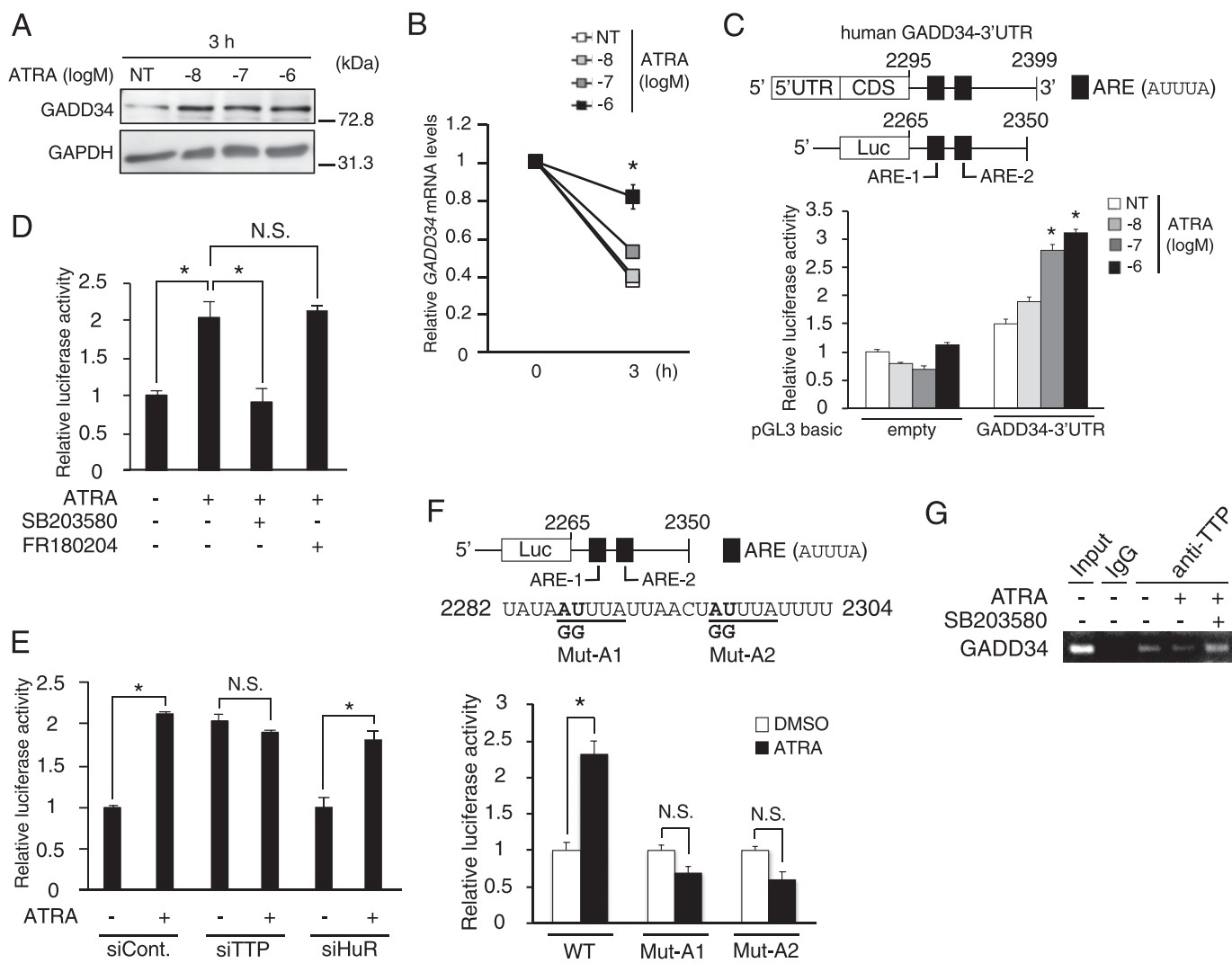

**Figure 6. Effects of all-trans retinoic acid (ATRA) on *GADD34* mRNA stability in C2C12 myoblasts.**
**(A)** Western blotting of GADD34 in C2C12 myoblasts incubated in the presence of 1 μM ATRA or DMSO (NT) as a vehicle for 3 h (*n* = 3–4). **(B)** The mRNA expression levels of *GADD34* in C2C12 myoblasts treated with the indicated concentrations of ATRA and 2.5 μg/ml actinomycin D (*n* = 3–4). *$P$ < 0.05 versus NT (one-way ANOVA with a Student–Newman post hoc test). **(C)** A schematic illustration of human *GADD34* mRNA sequence in the upper panels. pGL3-basic plasmid empty (control) or GADD34-3′UTR was transfected and incubated with the indicated concentrations of ATRA for 3 h in C2C12 myoblasts (*n* = 3–4). *$P$ < 0.05 versus NT (one-way ANOVA with a Student–Newman post hoc test). **(D, E)** Human *GADD34* mRNA 3′UTR reporter plasmid was transfected (E) with *TTP* siRNA (siTTP), *HuR* siRNA (siHuR), or control (siCont.) and incubated in the presence of (E) 1 μM ATRA or DMSO (D) and 10 μM SB203580 (p-p38 MAPK inhibitor) or FR180204 (p-ERK inhibitor) for 24 h (*n* = 3–4). *$P$ < 0.05 versus NT (DMSO) (one-way ANOVA with a Student–Newman post hoc test). **(F)** A schematic illustration of the mutated human *GADD34* mRNA sequence in the upper panels. Mut-A1 and Mut-A2 targeted the binding sites AREs for RNA-binding protein, respectively. Each human *GADD34* mRNA 3′UTR reporter plasmid (WT, Mut-A1, and Mut-A2) was transfected and incubated in the presence of 1 μM ATRA or DMSO as a vehicle for 24 h (*n* = 3–4). Data are presented as the mean ± SEM. *$P$ < 0.05 versus NT (DMSO) (one-way ANOVA with a Student–Newman post hoc test). **(G)** The mRNA expression levels of *GADD34* by RNA ChIP assay in C2C12 cells treated with 1 μM ATRA and 10 μM SB203580 for 24 h (*n* = 3–4). N.S., not significant. Similar results were obtained from independent experiments. Source data are available for this figure.

2 fast fibers, in myotubes. In addition, we confirmed that ATRA treatment decreased the protein expressions of total MYHC and MYHC2, but not MYHC1, in the GM of mice. Based on these results and reports, our findings may provide clues in relation to the unknown mechanisms of pathogenesis, such as muscle atrophy, in CKD patients.

Among the Six family of homeobox (Six1–6), Six1, Six2, and Six4 are expressed in myoblasts, Six1, Six2, and Six5 are expressed in myotubes (Kumar, 2009; Grand et al, 2012). MyoD reprogramming ability (whereby the MyoD expression turns extra muscle cells into a

muscle) is impaired in mouse embryonic fibroblasts in Six1/4 double mutant mice because Six1 and MyoD interact with the *Myogenin* gene promoter to differentiate muscle (Santolini et al, 2016). On the other hand, TLE3 down-regulates myogenic differentiation via the suppression of MyoD activity (Kokabu et al, 2017). Six1 requires the Eya family as the co-activator for transcriptional activation but the TLE family as the co-repressor for transcriptional suppression. In the present study, we suggested that—among the TLE family—*TLE3* is the most abundant gene in C2C12 cells and is an important co-repressor of Six1 for the regulation of the *GADD34*

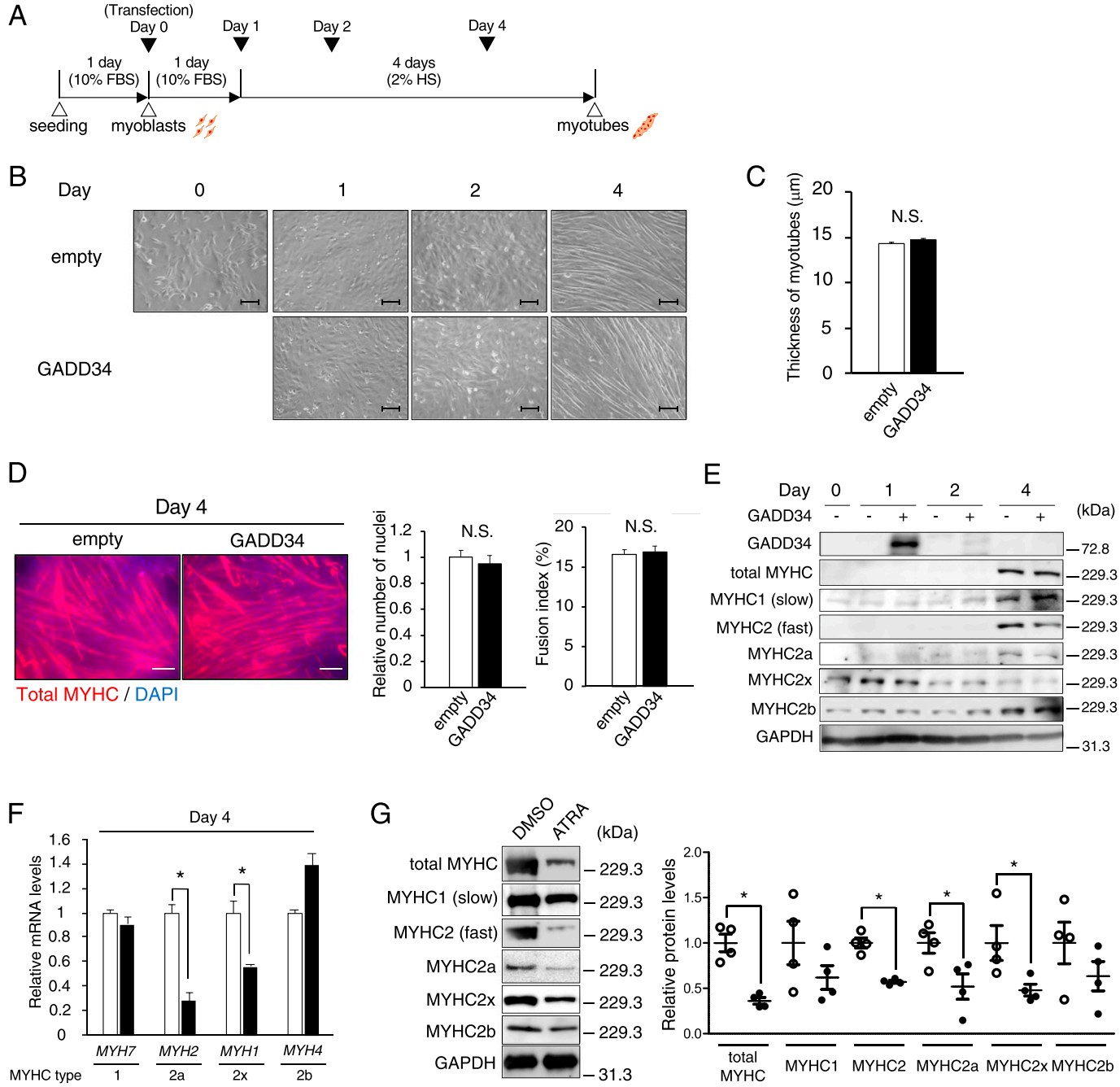

**Figure 7. Effects of the overexpression of GADD34 on the muscle fiber type in C2C12 myotubes.**
**(A)** A schematic illustration showing the experimental timeline in C2C12. **(B)** Representative images of C2C12 treated as indicated times (days 0, 1, 2, and 4). Scale bars = 100 μm. **(C)** The measurement of diameters of C2C12 myotubes on Day 4, as described in MATERIALS and METHODS (n = 100). **(D)** Immunofluorescence staining of total MYHC (green) and DAPI (blue) in C2C12 myotubes on Day 4, showed myotube formation in the C2C12 cells. Scale bars = 100 μm. Fusion index was as described in MATERIALS and METHODS (n = 100). **(E)** Western blotting of total MYHC, MYHC1 (Slow), MYHC2 (fast), MYHC2a, MYHC2x, and MYHC2b in C2C12 cells on each day (Days 0, 1, 2, and 4). **(F)** The mRNA expression levels of *MYH7*, *MYH2*, *MYH1*, and *MYH4* in Day 4 C2C12 cells (n = 3–4). *P < 0.05 (two-tailed unpaired *t* test). **(G)** Western blotting of total MYHC, MYHC1 (Slow), MYHC2 (fast), MYHC2a, MYHC2x, and MYHC2b in gastrocnemius muscle of all-trans retinoic acid-treated mice (n = 4). Data are presented as the mean ± SEM. *P < 0.05 (two-tailed unpaired *t* test). N.S., not significant. Source data are available for this figure.

gene transcriptional activity, which consequently changes the ratio of muscle fiber type. In addition, we confirmed that other undifferentiated muscle-specific transcriptional factors, Pax3 and Sox6, also suppress the *GADD34* gene promoter activity. Furthermore, we

also revealed that *GADD34* mRNA expression gradually increases with differentiation in C2C12 (Fig S4A). These findings suggest that the expression of GADD34 may constantly be kept low by not only Six1-TLE3 but also by undifferentiated muscle-specific transcriptional

factors. However, the reason why the expression of GADD34 is constantly low in myoblasts is unknown.

The expression of genes is determined by transcriptional regulation and post-transcriptional regulation via mRNA stability. The 3'UTR works as one of the regulatory components of mRNA degradation by promoting or inhibiting the deadenylation via interaction with RBP (Mayr, 2019). Although the classic target mRNAs that interact with RBPs for mRNA degradation are associated with inflammatory-related genes, a recent study revealed that other target mRNAs are related to cancer, apoptosis, and other conditions (Kaempfer, 2003; Lal et al, 2005; Schuster & Hsieh, 2019). However, UPR-related mRNA has not previously been reported as an ARE-dependent target for stabilization. TTP and HuR, the most famous RBP interacted with ARE (5'-AUUUA-3') on the 3'UTR, to induce mRNA degradation and stabilization, respectively. However, unlike TTP, HuR did not regulate GADD34 mRNA stability in the present study. The distinction between HuR and TTP binding has been reported to involve subtle content features: TTP: 5'-AUA[A/U/G/C][A/U]U[A/U]-3' and HuR: 5'-[A/C/U]UUUU[U/A/C][U/A]-3' (Bhandare et al, 2017). In other words, HuR strongly prefers U-rich sequences, whereas TTP prefers AU-rich with increasing A content. This may explain why HuR did not interact with the ARE sequences of GADD34 mRNA in the present study. ATRA activates the non-genomic p38 MAPK via extracellular RARα and RARγ (Tanoury et al, 2013). Furthermore, phosphorylated p38 inhibits the TTP interaction with ARE on target mRNA (Bhattacharyya et al, 2011). Our results indicated that ATRA stabilizes GADD34 mRNA by inhibiting the interaction between its mRNA and TTP via p38 MAPK in C2C12 myoblasts. Although it has been reported that ATRA regulates the stabilization of mRNA via interaction between apo-cellular retinoic acid-binding protein 2 (apo-CRABP2) and HuR (Vreeland et al, 2014), this is the first study to report that ATRA-dependent MAPK activation increases—via ARE—the mRNA stability of the 3'UTR of the target mRNA. Because p38 MAPK signaling regulates a large number of cellular processes, the mechanism underlying mRNA stabilization by ATRA may also be involved in many mRNAs other than GADD34 mRNA.

Although the Six family genes, especially Six1, are expressed in multiple organs during mammalian development, their expression in global tissues decreases as the individual grows to adulthood (Kumar, 2009). Six1 also controls muscle physiology not only in embryogenesis but also in the adult (Maire et al, 2020). In soleus, Six1 deficiency reduced MYHC2a fiber in 3-wk-old C57BL6 mice and caused the complete disappearance of the expression of MYHC2a fiber in 12-wk-old C57BL6 mice (Sakakibara et al, 2016). These results imply that Six1 regulates fast fiber type acquisition and maintenance in adult mice. Because our study revealed that GADD34, which is down-regulated by Six1, suppressed the MYHC2a expression in C2C12 myotubes, GADD34 may also mediate the regulation of the expression of MYHC2a by Six1 in vivo. Aside from this, GADD34 induces cellular senescence via the regulation of the p21 expression in MEF and 7EJ-Ras cells (Minami et al, 2007). Cellular senescence, a permanent state of replicative arrest in otherwise proliferating cells, is a hallmark of aging and has been linked to aging-related diseases (Childs et al, 2015). Because the plasma levels of vitamin A (retinol or ATRA) increase with aging (Gueguen et al, 2005), the increase of ATRA that occurs with aging may accelerate

cellular senescence in the muscle via the induction of GADD34, resulting in muscle atrophy in elderly patients. It has been reported that aged mice exhibit an increase of slow muscle fibers but a decrease of fast muscle fibers (Shang et al, 2020). These results are in line with the effects of GADD34 on myotubes that were observed in our study. Because we also revealed that ATRA mediates muscle atrophy via decreasing both MYHC types in mice, ATRA-induced GADD34 protein may contribute to the acceleration of decreasing type 2 MYHC expressions.

In conclusion, as shown in Fig 8, our findings revealed that the up-regulation of the GADD34 gene expression by ATRA contributes to muscle fiber type change in skeletal muscle. Furthermore, ATRA and its receptors can increase the transcriptional activity of the GADD34 gene by blocking the interaction of Six1-TLE3 with the GADD34 gene promoter through the ERK signal in myoblasts and myotubes. ATRA also stabilizes GADD34 mRNA by inhibiting the interaction of p38-TTP with the 3'UTR of GADD34 mRNA in myoblasts.

## Materials and Methods

### Chemicals and reagents

ATRA, DMSO, high-glucose DMEM, FBS, FR180204, anti-GAPDH antibody (G8795), anti-Six1 antibody (HPA001893), anti-Laminin antibody (L9393), goat anti-mouse IgG (H + L)-HRP conjugate, goat anti-rabbit IgG (H + L)-HRP conjugate, 4',6-diamidino-2-phenylindole (DAPI; D9542), MISSION siRNA oligos were purchased from Sigma-Aldrich. Buprenorphine hydrochloride was purchased from Otsuka Pharmaceutical Co., Ltd.. Pentobarbital sodium salt was purchased from Tokyo Kasei Co., Ltd. Anti-phosphorylated eIF2α (p-eIF2α) (#9721, Ser51), anti-eIF2α (#9722), anti-ATF4 (#11815), anti-CHOP (#2895), anti-TTP (#71632), anti-phosphorylated ERK (p-ERK) (#9101, Thr202/Try204), and anti-ERK (#9102) antibody were purchased from Cell Signaling Technology. Anti-GADD34 (sc-373815), anti-phosphorylated p38 (p-p38) (sc-166182 and Tyr182), anti-p-38 (sc-7972), and anti-HuR (sc-5261) antibody were purchased from Santa Cruz Biotechnology. Anti-GADD34 antibody (10449-1-AP) was purchased from Proteintech. Anti-total MYHC (MF20), anti-MYHC1 (BA-D5), anti-MYHC2 (F59), anti-MYHC2a (SC-71), anti-MYHC2x (6H1), and anti-MYHC2b (BF-F3) antibody were purchased from Developmental Studies Hybridoma Bank. Anti-puromycin antibody was purchased from Cosmo Bio. [γ-32P] ATP was purchased from ICN. T4 polynucleotide kinase and TransIT-LT1 Reagent were purchased from TAKARA. SB203580 was purchased from Adipogen Life Science.

Alexa Fluor 488 goat anti-rabbit IgG, Alexa Fluor 546 goat anti-mouse IgG, human recombinant TNFα were purchased from Invitrogen. Puromycin was purchased from Funakoshi. Low-glucose DMEM and Chemi-Lumi One Super were purchased from Nacalai Tesque. QIAzol Lysis Reagent was purchased from QIAGEN. TOYOBO KOD one PCR Master Mix Blue was purchased from TOYOBO. RNA ChIP-IT was purchased from Active Motif. Collagenase I was purchased from Worthington Biochemical Corporation. Horse serum (HS) was purchased from Moregate Biotech. Aqua Poly/Mount was purchased from Polysciences Inc.

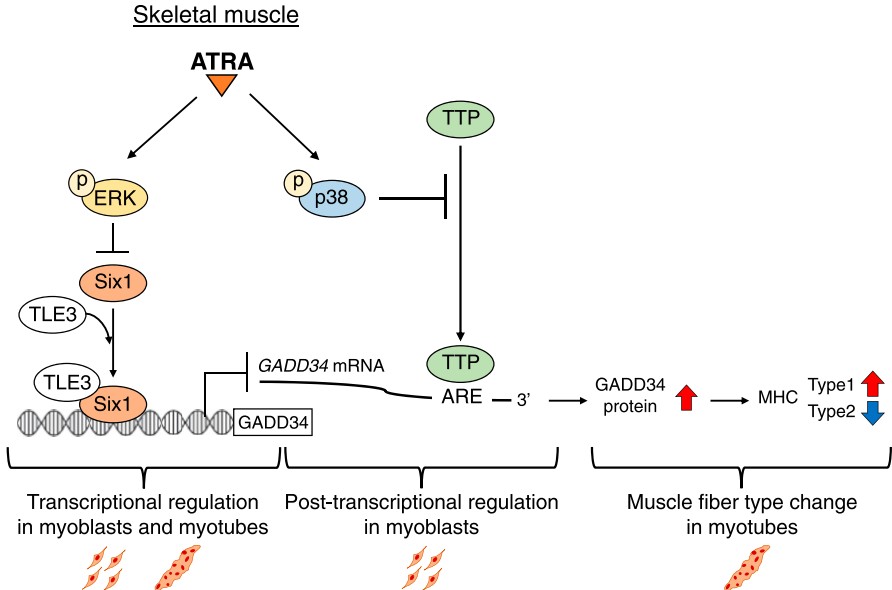

Skeletal muscle

**Figure 8.** Schematic illustration of muscle-specific GADD34 induction through all-trans retinoic acid (ATRA)–dependent transcriptional regulation and post-transcriptional regulation.

ATRA decreases the expression of Six1, which decreases the transcriptional activity of *GADD34* with TLE3, via ERK phosphorylation in myoblasts and myotubes. ATRA suppresses TTP-induced *GADD34* mRNA degradation through the binding of ARE on the 3′UTR of the *GADD34* mRNA in a p38-dependent manner in myoblasts. These events increase the expression of GADD34 proteins, resulting in the change in the type of myosin heavy chain in myotubes.

## Animal experiments

The animal work took place in Division for Animal Research and Genetic Engineering Support Center for Advanced Medical Sciences, Institute of Biomedical Sciences, Tokushima University Graduate School. 8-wk-old male C57BL/6J mice were purchased from Japan SLC and individually caged in a climate-controlled room (22°C ± 2°C) with a 12-h light–dark cycle. These mice were randomly divided into two groups and a total of 1 mg/kg body weight of ATRA or 1% DMSO (control) prepared in sterile saline was intraperitoneally administered, then the mice were euthanized 24 h later. Before euthnasia, mice were anesthetized with a total of 0.1 mg/kg body weight of buprenorphine hydrochloride and a total of 50 mg/kg body weight of pentobarbital sodium salt, and tissues were removed. The present study was approved by the Animal Experimentation Committee of Tokushima University School of Medicine (animal ethical clearance No. T30-66) and was carried out in accordance with guidelines for the Animal Care and use Committee of Tokushima University School of Medicine.

## Primary culture

Single myofiber cultures were established from GM muscle after digestion with collagenase I and trituration as described (Pichavant & Pavlath, 2014). Suspended myofibers were cultured in 60 mm horse serum-coated plates in high-glucose DMEM containing 10% FBS, 100 units/ml penicillin and 100 $\mu$g/ml streptomycin at 37°C with 5% $CO_2$ for 24 h.

## Cell culture

C2C12 myoblast cells were cultured as described previously (Niida et al, 2020). Briefly, C2C12 myoblast cells were cultured in high-glucose DMEM containing 10% FBS, 100 units/ml penicillin and 100 $\mu$g/ml streptomycin at 37°C with 5% $CO_2$. At 100% confluence, C2C12 myoblast cells were fused by shifting the medium to DMEM containing 2% horse serum (HS) for differentiation. These cells were maintained in 2% HS/high-glucose DMEM (differentiation medium) for 4 d before experiments. After the differentiation, we treated ATRA or DMSO as vehicle control in 2% HS/high- or low-glucose DMEM each for 24 h as shown in Fig 1E. HEK293 cells were cultured in DMEM containing 10% FBS, 100 units/ml penicillin, and 100 $\mu$g/ml streptomycin at 37°C with 5% $CO_2$.

## Western blotting

Protein samples were heated to 95°C for 5 min in sample buffer in the presence of 5% 2-mercaptoethanol and subjected to SDS–PAGE. The separated proteins were transferred by electrophoresis to polyvinylidene difluoride transfer membranes (Immobilon-P; Millipore). The membranes were treated with diluted affinity-purified anti-p-eIF2α (p-eIF2α) (1:1,000), anti-eIF2α (1:1,000), anti-ATF4 (1:3,000), anti-CHOP (1:1,000), anti-GADD34 (10449-1-AP) (1:1,000), anti-Six1 (1:1,000), anti-TTP (1:1,000), anti-HuR (1:1,000), anti-p-p38 (p-p38) (1:1,000), anti-p38 (1:1,000), anti-p-ERK (p-ERK) (1:1,000), anti-ERK (1:1,000), anti-total MYHC (1:1,000), anti-MYHC1 (1:1,000), anti-MYHC2 (1:1,000), anti-MYHC2a (1:1,000), anti-MYHC2x (1:500), anti-MYHC2b (1:500), and anti-puromycin (1:2,000) antibody. Mouse anti-GAPDH monoclonal antibody was used as an internal control. Goat anti-mouse IgG (H + L)–HRP conjugate (1:3,000), and Goat anti-rabbit IgG (H + L)-HRP conjugate (1:3,000) was used as the secondary antibody, and signals were detected using the Chemi-Lumi One Super.

## Immunofluorescence staining

For determination of Laminin and GADD34 expression, GM was embedded in cryomatrix and quickly frozen in isopentane cooled with liquid nitrogen. Cryostat sections (10 $\mu$m) were washed in PBS,

permeabilized with 0.1% Triton X-100 for 20 min, and left for 1 h in blocking solution (1× PBS, 1.5% goat serum, 0.1% Triton X-100). Rabbit poly-clonal antibodies directed against Laminin (1:100), and mouse monoclonal antibodies against GADD34 (sc-373815) (1:100) were applied overnight at 4°C to the treated sections. The next day, after three washes with 1× PBS containing 0.05% Tween-20, sections were incubated for 1 h with appropriate fluorescent secondary antibodies (Alexa Fluor 488 goat anti-rabbit IgG 1/1,000 dilution, Alexa Fluor 546 goat anti-mouse IgG 1/1,000 dilution) and 0.5 $\mu$g/ml DAPI. After three washes with 1× PBS containing 0.05% Tween-20, samples were mounted in Aqua Poly/Mount.

For determination of the fusion index, the C2C12 cells were fixed in 4% paraformaldehyde for 20 min at room temperature. Mouse monoclonal antibodies against total MYHC (1:1,000) were applied overnight at 4°C to the treated sections. The next day, after three washes with 1× PBS containing 0.05% Tween-20, sections were incubated for 1 h with appropriate fluorescent secondary antibody (Alexa Fluor 546 goat anti-mouse IgG 1/1,000 dilution) and 0.5 $\mu$g/ml DAPI. After three washes with 1× PBS containing 0.05% Tween-20, differentiated myotubes in a specific microscopic field were observed under ×20 magnification. Either the total number of nuclei or the number of nuclei within MyHC-positive myotubes was counted in five fields/groups. The fusion index was calculated as follows: (%) = (number of nuclei within MyHC-stained myotubes/total number of nuclei) × 100.

## Real-time quantitative PCR

Total RNA was isolated from kidney, liver, EA, GM, femur, C2C12 myoblasts, C2C12 myotubes, and HEK293 cells using an QIAzol Lysis Reagent. Real-time quantitative PCR (real-time qPCR) assays were performed using an Applied Biosystems StepOne qPCR instrument. In brief, the cDNA was synthesized from 1 $\mu$g of total RNA using a reverse transcriptase kit (Invitrogen) with an oligo-dT primer. After cDNA synthesis, real-time qPCR was performed in 5 $\mu$l of SYBR Green PCR master mix using a real-time PCR system (Applied Biosystems). Amplification products were then analyzed by a melting curve, which confirmed the presence of a single PCR product in all reactions (apart from negative controls). The quantification of given genes was expressed as the mRNA level normalized to a ribosomal 18S housekeeping gene using the ΔΔCt method. Quantitative expression values were calculated from an absolute standard curve method using the plasmid template for each target gene. The primer sequences used for real-time qPCR analysis are shown in Table S1.

## Reporter plasmid construction

The promoter fragment of luciferase reporter plasmid pGADD34-0.5k generated by Eurofins Japan was subcloned into a pGL3 vector (Promega) by restriction enzyme cutting site KpnI/HindIII. Reporter plasmids pGADD34-0.05k and pGADD34-0.5kΔ-0.05k were cloned by digestion of pGADD34-0.5k using TaqI restriction enzyme. Luciferase reporter plasmids GADD34-3'UTR was constructed by PCR amplification of human genomic cDNA as a template using gene-specific primers (Table S2). These products were subcloned into a pGL3 vector. Deleted reporter plasmid pGADD34-0.3k was cloned by the digestion of pGADD34-0.5k using DpnI/HindIII. Deleted reporter

plasmids pGADD34-0.16k, pGADD34-0.13k, and mutated reporter plasmids pGADD34-0.16k-Mut-MEF3-1 (Mut-M1), pGADD34-0.16k-Mut-MEF3-2 (Mut-M2), pGADD34-0.16k-Mut-MEF3-3 (Mut-M3), pGADD34-0.05k-Mut-1 (Mut-1), pGADD34-0.16k-Mut-2 (Mut-2), GADD34-3'UTR-Mut-ARE-1 (Mut-A1), and GADD34-3'UTR-Mut-ARE-2 (Mut-A2) were constructed with TOYOBO KOD one PCR Master Mix Blue using the oligonucleotides shown in Table S2. The $\beta$-galactosidase expression vector pCMV-$\beta$ (CLONTECH) was used as an internal control. Each plasmid was purified with a FavorPrep Plasmid DNA Extraction Midi Kit (Favorgen).

## Transfection and luciferase assay

Mouse Six1, Pax3, and Sox6 expression vector (pCR3-Six1, pCR3-Pax3, and pCR3-Sox6) were kindly provided by Dr. P Maire (Santolini et al, 2016). Human GADD34 expression vector (pRP-Neo-CMV-hPPP1R15A) was designed by Vector Builder. Cells were transfected by TransIT-LT1 Reagent, then treated with several concentrations of ATRA, SB203580, FR180204, TNF$\alpha$, or DMSO as vehicle control for an additional 16 h. A luciferase assay was performed as described previously (Masuda et al, 2010).

## RNAi experiments

C2C12 myoblasts were transfected with siRNA directed against *ATF4* (SASI_Mm02_00316863), *Six1* (SASI_Mm01_00198104), *TLE1* (SASI_Mm01_00069933), *TLE3* (SASI_Mm02_00300046), *TLE5* (SASI_Mm01_00139428), *TTP* (SASI_Mm01_00178605), *HuR* (SASI_Mm02_00318722), or negative control (SIC001) using TransIT-LT1 Reagent, according to the manufacturer's instructions.

## EMSA

EMSA was performed as described previously (Masuda et al, 2020). Double-stranded nucleotides for GADD34-MEF3 and GADD34-MEF3-2-Mutant (GADD34-Mut-M2) were synthesized (Table S3). Purified DNA fragments were radiolabeled with [$\gamma$-32P] ATP (110 TBq/mmol) using T4 polynucleotide kinase. Nuclear extracts (pCR3-Six1) were prepared as described previously (Masuda et al, 2010). Briefly, the C2C12 myoblast cells were cultured in 10-cm dishes to 60% confluence and transfected with pCR3-Six1 or treated with 1 $\mu$M ATRA. Prepared nuclear extracts (15 $\mu$g) were incubated with the radiolabeled probe in binding buffer (10 mM [Tris–HCl], pH 7.5, 1 mM DTT, 1 mM EDTA, 10% glycerol, 1 mM MgCl2, 0.25 mg/ml bovine serum albumin, 2.5 $\mu$g/ml salmon sperm DNA, and 2 $\mu$g poly(dI-dC)) in a final volume of 20 $\mu$l for 30 min at room temperature. The specificity of the binding reaction was determined with a 100-fold molar excess of the indicated cold competitor oligonucleotide. The reaction mixture was then subjected to electrophoresis on a 5% polyacrylamide gel with 0.25 × TBE running buffer for 2 h at 150 V. The gel was dried and analyzed with an image scanner (FLA-9000 Starion).

## ChIP assay

Chromatin immunoprecipitations were performed essentially as described previously (Imbriano et al, 2005). Briefly, exponentially growing C2C12 cells were washed with phosphate-buffered saline and incubated for 10 min with 1% formaldehyde. The cross-linked

material was broken with a Dounce pestle, and chromatin-containing pellets were sonicated to get fragments of an average length of 500 bp. C2C12 cells were immunoprecipitated for 4 h at 4°C with 3 $\mu$g of anti-Six1 antibody. Semi-quantitative PCRs were performed in the linear range of each amplification product; ChIP-PCR primers used for these experiments are listed in Table S4.

### RNA ChIP assay

Using the RNA ChIP-IT, an RNA ChIP assay was performed according to the manufacturer's instructions. Briefly, the C2C12 myoblast cells were cultured to 80% confluence in 10-cm dishes and treated with 1 $\mu$M ATRA. The C2C12 cells were collected and lysed in lysis buffer. The cell extract was prepared and incubated with RNA ChIP buffer pre-conjugated with TTP antibodies or control mouse IgG at 4°C for 16 h. The complexes were treated with Proteinase K for 1 h at 45°C and 1.5 h at 65°C. Immunoprecipitated RNA in the precipitates was purified using QIAzol Lysis reagent and analyzed for GADD34 by reverse transcription-PCR (RT-PCR).

### Measurement of C2C12 diameters

The myotube diameters were determined as previously reported (Niida et al, 2020). Under a fluorescence microscope (BIOREVO BZ-9000; Keyence), three photographs per cell-culture well were obtained in a high-power field. We measured the diameter at the middle portion of the myotube with the built-in BZ-II analyzer software program. We measured the diameters of 100 myotubes/group.

### Surface sensing of translation (SUnSET) assay in C2C12 cells

C2C12 was incubated with or without 1 $\mu$M puromycin for 30 min before collecting the cells and washed with PBS as previously described (Lim et al, 2017). Puromycin-labeled proteins were detected by Western blotting, as shown above.

### Statistical analysis

Data were collected from more than two independent experiments and were reported as the mean and SEM. The statistical analysis for two-group comparison was performed using a two-tailed $t$ test or one-way ANOVA with Tukey–Kramer post hoc test for multigroup comparison. All data analyses were performed using the GraphPad Prism 5 software program (GraphPad Software). $P$-values of <0.05 were considered to indicate statistical significance.

# Supplementary Information

# Acknowledgements

This work was supported by JSPS KAKENHI Grant Numbers 17K19910, 17H05061, 20K21761, 21H03359 (to M Masuda), 19K22811, and 19H04053 (to Y Taketani). We thank K Sato, K Sasaki, R Kimura, Y Nobe, S Aoyagi, and A Yamada (Department of Clinical Nutrition and Food Management, Institute of Biomedical Sciences, Tokushima University Graduate School, Tokushima, Japan) for technical assistance. We also thank Support Center for Advanced Medical Sciences, Tokushima University Graduate School of Biomedical Sciences, for technical assistance.

## Author Contributions

Y Adachi: conceptualization, resources, data curation, software, formal analysis, validation, investigation, visualization, methodology, and writing—original draft.
M Masuda: conceptualization, supervision, funding acquisition, validation, project administration, and writing—original draft, review, and editing.
I Sakakibara: formal analysis, supervision, validation, and methodology.
T Uchida: formal analysis, validation, investigation, and methodology.
Y Niida: validation and methodology.
Y Mori: validation and investigation.
Y Kamei: validation and investigation.
Y Okumura: validation and investigation.
H Ohminami: formal analysis, validation, and methodology.
K Ohnishi: data curation, formal analysis, validation, and methodology.
H Yamanaka-Okumura: data curation, formal analysis, validation, and methodology.
T Nikawa: validation, visualization, and methodology.
Y Taketani: supervision, funding acquisition, validation, visualization, project administration, and writing—original draft, review, and editing.

## Conflict of Interest Statement

The authors declare that they have no conflict of interest.

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
