## [Reviewer comments · Life Science Alliance]

Life Science Alliance

All-trans retinoic acid changes muscle fiber type via increasing GADD34 dependent on MAPK signal

Yuichiro Adachi, Masashi Masuda, Iori Sakakibara, Takayuki Uchida, Yuki Niida, Yuki Mori, Yuki Kamei, Yosuke Okumura, Hirokazu Ohminami, Kohta Ohnishi, Hisami Yamanaka-Okumura, Takeshi Nikawa, and Yutaka Taketani

DOI: <https://doi.org/10.26508/lsa.202101345>

Corresponding author(s): Masashi Masuda, Tokushima University Graduate School

Review Timeline:	Submission Date:	2021-12-21
	Editorial Decision:	2021-12-23
	Revision Received:	2022-02-16
	Editorial Decision:	2022-03-07
	Revision Received:	2022-03-09
	Accepted:	2022-03-11

Transaction Report:

Please note that the manuscript was reviewed at Review Commons and these reports were taken into account in the decision-making process at Life Science Alliance.

Review #1

****Summary:****

The authors investigate the regulation of GADD34 in mouse myoblasts downstream of retinoic acid treatment. Using mechanistic experiments in C2C12 myoblasts, the authors provide evidence that all-trans retinoic acid (ATRA) treatment results in upregulation of GADD34 expression in gastrocnemius muscle and determine, using C2C12 cells that this stimulation confined to myoblast and not differentiated myotubes. They attribute the effect of ATRA to indirect regulation of Six1 binding to a MEF3 response element in the proximal promoter of GADD34, ultimately concluding that ATRA derepresses GADD34 expression in myoblasts through downregulation of SIX1 expression. The authors provide further evidence that the GADD34 mRNA is regulated by a 3'-UTR which results in increased instability and rapid mRNA degradation that is stabilized in response to ATRA treatment. Both mechanisms are proposed to act to increase GADD34 expression in myoblasts in response to retinoic acid treatment. Finally, the authors investigate the biological impact of GADD34 stimulation by ATRA during myogenic differentiation, and conclude that while differentiation is not affected by ectopic expression of GADD34, changes in myosin heavy chain expression were noted.

****Major comments:****

The data presented suggest an interesting two-fold mechanism for the regulation of GADD34 by all trans-retinoic acid. Given the important impact of ATRA on myogenic differentiation, it is an interesting area of investigation and could contribute to our understanding of the complex mechanisms by which RA influences cell differentiation. However, a major limitation of the work presented is that it is performed entirely in C2C12 myoblasts. While these cells are a useful model, they do not completely recapitulate in vivo muscle regeneration and there are many reported differences between C2C12 and primary myoblasts. As such, this field of study has largely moved away from using this model without validation in vivo or in primary myoblasts.

The mechanistic approach used is systematic and relies on techniques that provide detailed understanding. However, the scope of the investigation is narrow and does not completely describe the mechanism, resulting in many new questions and unclear biological importance.

The experiments are, in general, rigorously performed and presented. However, there are several instances of missing controls that would better support the conclusions made and not all conclusions are well-supported by the data presented. There are numerous concerns related to the methods that need to be addressed in order to fully assess the data presented in this manuscript.

In particular:

1. Figure 1. This figure demonstrates that the upregulation of Gadd34 mRNA by all-trans retinoic acid (ATRA) is unique to skeletal muscle and the authors provide evidence, using the C2C12 model that the effect is restricted to myoblasts (undifferentiated cells) and not myotubes (Fig. 1D). This result is confusing as it is expected that the sample used in Fig 1A was derived from primarily myofibers, with a small component from muscle satellite cells and other non-muscle cell types. As such, the findings that in C2C12 the effect is limited to myoblasts and absent in myotubes is surprising. To clarify, IHC of muscle sections for GADD34 expression following RA treatment would help identify which cell population is responsible for the upregulation. In parallel, it is necessary to validate the findings from C2C12 cells in primary myoblasts, as C2C12 do not entirely recapitulate myogenic differentiation seen in primary cells or in vivo.

Figure 1F. While the magnitude of ATF4 knockdown is stated in the results section on page 7, the validation should be included in the figure. In addition, treatment of the siRNA for ATF4 cells should also be treated with vehicle and ATRA for completion.

Figure 2A. In this panel, C2C12 in the absence of forced expression of the RARs/RXR does not stimulate transcription from the minimal Gadd34 promoter. C2C12 do express RARs and RXR and their myogenic differentiation can be stimulated by ATRA. If the effect is limited to C2C12 cells that express ectopic receptors, the importance of the finding is unclear, especially given that the baseline transcription increases with RAR

addition. Validation in primary cells is required, and a more complete evaluation of the regulatory regions controlling GADD34 should be performed. While it is clear that the authors wish to advance a model that the RARs inhibit Six1 levels and therefore derepress GADD34 expression, which may be an important regulatory mechanism, the literature suggests that PPAR γ can stimulate GADD34 expression directly when a promoter construct of approximately 2kb in length is used (reference: PMID: 12881480). The authors should justify the minimal construct studied and why larger constructs were excluded. The authors should also examine published data sets for RA treatment that may support or refute the direct regulation of the GADD34 regulatory region in myoblasts. ChIP assay, especially in primary myoblasts, would support the model that RA does or does not interact with the regulatory regions.

Figure 3. The authors propose that a single MEF3 element is responsible for transducing the effects of RA on the GADD34 promoter construct. This is supported by mutational analysis, but the mutation for site MEF3-2 overlaps both the MEF3-1 and the MEF3-3 sites. Confirmation that the mutations prevent binding of the individual sites without impacting neighbouring sites is not provided but is necessary. Is MEF3-1 and MEF3-3 binding affected by the mutations in the MEF3-2 oligo? For panel 3H, the controls must be included if this data is to be interpreted.

Figure 4. Panel 4F should include both ATRA treatment and ectopic Six1 to fully support the mechanism proposed. ChIP should be completed to support the idea that Six1 occupancy of the Mef3-2 site is affected by ATRA treatment.

Figure 5. In panel A, it is unclear why the complete data including error is not provided. In panel E, samples that receive Six1 should also be treated with RA. Further, siRNA against Six1 could provide useful information instead of relying uniquely on overexpression systems.

Figure 6. The labelling and legend for panel A makes it impossible to distinguish from Figure 1E, though the text suggest that this blot is from samples treated for only 3 hours. The data presented in panels C-F are problematic. The pGL3-basic vector does not contain a promoter and thus is not expected to result in the production of luciferase. However, the authors use this vector to assess the importance of the GADD34 3'-UTR on mRNA stability in this context. There is concern that the data generated is captured from signal that is not in a range that allows for accurate interpretation. This should be repeated using a vector containing a minimal promoter that is demonstrated not to be affected by ATRA treatment.

Figure 7. This figure explores the biological importance of the described mechanism by forcing expression of GADD34 in C2C2 myoblasts. The results indicate that there is no difference in the ability of these myoblasts to differentiate, though there are several important problems with the data as presented. Since the authors demonstrate that the upregulation of GADD34 occurs in proliferating myoblasts and not myotubes, the authors should characterize the function of these cells more completely. Is proliferation affected +/-RA and +/-GADD34? Loss of function experiments would also be very useful. Does knockdown of Gadd34 have an impact on myogenic differentiation? In addition, Six1 is known to be expressed in myoblasts and its expression decreases with differentiation (<https://academic.oup.com/nar/article/38/20/6857/1309345>). If the model is correct, do Gadd34 levels surge in myotubes? A time course of Gadd34 expression would be beneficial for understanding.

Figure 7B. The assessment of myogenic differentiation is normally achieved by performing myosin heavy chain IIF and calculating the percentage of nuclei in MyHC+ cells as compared to total nuclei as well as myotube fusion (number of nuclei per myotube). The assessment of myotube thickness is an unreliable measure of differentiation. Further, while there appears to be efficient differentiation based on the images in panel B, the trajectory of Pax7 expression is not as expected for controls and the authors should comment on this. Most importantly, the successful overexpression of GADD34 is not shown in the main figure, and without this important control, it is impossible to assess this data. In panel E, the expression for GADD34 is missing, but is a critical control for assessing the data presented.

Figure 7F. The legend is missing from this panel and the mRNA levels of Gadd34 are missing but required.

Taken together, there are significant revisions required to render the conclusions convincing. Two mechanisms are proposed for the regulation of GADD34 expression in myoblasts, however neither mechanism is explored in detail, rendering the conclusions preliminary in nature and their generalizability, and therefore impact, small.

The requested experiments are in line with the scope of the manuscript and are reasonable, especially given the limited model used by the authors.

Overall, the data are well-presented and the methods are clear. Adequate replicates and analysis were performed.

****Minor comments:****

1. The Abstract needs to be re-written for clarity. In its present form, while the main ideas are stated, the sentence structure prevents understanding. In particular lines 15-17 are problematic.
2. Additional information about the known mechanisms regulating Gadd34 expression should be added to the Introduction.

The area of research is important and has the potential to contribute to the mechanistic understanding of retinoic acid receptor function during myoblast differentiation. Interesting data is provided, however, the story feels at times incomplete and requiring further information to significantly advance the field. The data presented herein is derived primarily from one cell line that has been shown to not be an entirely reliable model of myogenic differentiation. This coupled with an over-reliance on overexpression systems, lack of controls for some experiments, and concerning experimental design in Figure 6, prevents significant advancement of knowledge. The authors fail to place their work in the context of the literature, including the discovery of RXR binding to the GADD34 promoter and the expression of Six1, including excellent datasets that could support binding to the GADD34 promoter region in myoblasts. A more complete alignment with the models proposed and the literature would improve the manuscript overall, though more use of primary cells and in vivo models would be required to rise this contribution to the level expected by the journal.

In general the work presented is geared towards a specialty audience that is interested in mechanism of retinoic acid action in myoblasts.

As a reviewer, my research aims to understand the transcriptional mechanisms regulating myogenic differentiation. I am an expert in nuclear receptor biology and retinoic acid signaling, and can appreciate the importance of the mechanistic detail proposed. I also, however, recognize that the work does not represent the cutting edge of the field in terms of techniques used, approaches, or use of available datasets.

Review #2

This manuscript by Adachi investigated the mechanisms by which all-trans retinoic acid (ATRA) induces the expression of Gadd34 gene expression in skeletal muscles. Mainly by cell culture-based assays, Adachi et al showed that ATRA induced Gadd34 gene expression in undifferentiated myoblasts by two distinct mechanisms: one is to reduce the expression of Six1, a negative regulator for the transcription of Gadd34 gene; and the other is to decrease the binding of tristetraprolin (TPP), an RNA-destabilizing protein, to 3'UTR of Gadd34 mRNA resulting in its stabilization. Overexpression of Gadd34 in C2C12 myoblasts did not affect differentiation but slightly reduced the expression of two fast-type myosin heavy chain proteins.

****Major points:****

1. Gadd34 gene was shown to be induced in the mouse gastrocnemius muscles (mainly composed of differentiated myofibers) (Figure 1A). However, in cultured C2C12 myoblasts, Gadd34 gene was shown to be only induced in undifferentiated myoblasts but not differentiated myotubes (Figure 1D). How do you explain this obvious discrepancy?
2. Although mouse C2C12 cells were extensively used for mechanistic studies in this manuscript, it is counter-intuitive that authors chose to use human Gadd34 promoter for detailed transcriptional analysis in this manuscript. It is never mentioned whether some of the key cis-elements identified in this manuscript (e.g., the MEF3 site, or the ARE element in 3'UTR) are conserved in the mouse Gadd34 promoter or not.
3. Although overexpression of Gadd34 in C2C12 myoblasts slightly reduced the expression of two fast-type myosin heavy chains, it is premature to claim that elevated Gadd34 promotes muscle fiber type switch. More rigorous assays (including more in vivo assays) are required in order to draw such a conclusion.

****Minor points:****

1. Figure 3G: the exogenously transfected Six1 was used in EMSA to prove that the MEF3-2 site of the Gadd34

promoter was bound by Six1. As Gadd34 gene was normally repressed in C2C12 myoblasts in the absence of ATRA, one would expect that the endogenous Six1 also binds to the Gadd34-MEF3 probe. It would be more convincing to use the endogenous Six1 in such EMSA experiments.

2. Figure 7D: In cultured C2C12 myoblasts, the endogenous Pax7 mRNA is known to be expressed at very low levels (Seale et al., *Cell*, 2000). Moreover, Pax7 gene expression further goes down upon differentiation (Zammit, et al., *J Cell Sci*, 2006). Therefore, it is counter-intuitive to see a gradual increase in Pax7 mRNA upon myoblast differentiation.

Overall, the cell culture-based assays in this manuscript were generally solid and well controlled. The effect of ATRA on Gadd34 gene induction was also clear. However, without solid *in vivo* data and a clear link of the central research question with a well-defined developmental process or disease setting, the impact and the significance of such studies are not high.

December 23, 2021

Re: Life Science Alliance manuscript #LSA-2021-01345

Dr. Masashi Masuda
Tokushima University Graduate School
Clinical Nutrition and Food Management
Japan

Dear Dr. Masuda,

Thank you for submitting your manuscript entitled "All-trans retinoic acid induces GADD34 gene expression via transcriptional regulation by Six1-TLE3 and post-transcriptional regulation by p38-TTP in skeletal muscle" to Life Science Alliance. We invite you to re-submit the manuscript, revised according to your Revision Plan.

Thank you for this interesting contribution to Life Science Alliance. We are looking forward to receiving your revised manuscript.

Sincerely,

B. MANUSCRIPT ORGANIZATION AND FORMATTING:

The point-by-point response to reviewer comments

Dear Prof. Eric Sawey, Executive Editor, Life Science Alliance

Manuscript No.: LSA-2021-01345,

“All-trans retinoic acid induces GADD34 gene expression via transcriptional regulation by Six1-TLE3 and post-transcriptional regulation by p38-TTP in skeletal muscle”

Corresponding author: Masashi Masuda

We wish to express our appreciation to the reviewers for their insightful comments on our paper. The comments have helped us significantly improve the paper. As indicated in the responses that follow, we have taken all these comments and suggestions into account in the revised version of our paper.

Reviewer #1:

Comments

1. Figure 1. This figure demonstrates that the upregulation of Gadd34 mRNA by all-trans retinoic acid (ATRA) is unique to skeletal muscle and the authors provide evidence, using the C2C12 model that the effect is restricted to myoblasts (undifferentiated cells) and not myotubes (Fig. 1D). This result is confusing as it is expected that the sample used in Fig 1A was derived from primarily myofibers, with a small component from muscle satellite cells and other non-muscle cell types. As such, the findings that in C2C12 the effect is limited to myoblasts and absent in myotubes is surprising. To clarify, IHC of muscle sections for GADD34 expression following RA treatment would help identify which cell population is responsible for the upregulation. In parallel, it is necessary to validate the findings from C2C12 cells in primary myoblasts, as C2C12 do not entirely recapitulate myogenic differentiation seen in primary cells or in vivo.

Response: Thank you so much for your valuable suggestion. We checked ATRA-induced GADD34 localizes in myofibers mainly in gastrocnemius muscle (GM) by analyzing IHC in Fig 1C. We also revealed that ATRA increased GADD34 mRNA expression in isolated primary myofiber in Fig 1D. Therefore, we hypothesize that ATRA increases the expression of GADD34 in not only myoblasts but also myotubes. Recent reports show that there are cases in which low-glucose DMEM is better to get a similar signal with in vivo study than high-glucose DMEM, which is the most common medium to culture C2C12 classically. We confirmed that ATRA increased GADD34 mRNA expression in low-glucose DMEM cultured C2C12 myotubes in Fig 1F. To investigate the reason why the sensitivity of ATRA to the GADD34 gene promoter activity is different in the status of C2C12 cells, we checked the effects of ATRA on MAPK signal in GM and C2C12 myotubes cultured with high- or low-glucose. As a results, ATRA-induced phosphorylation of p38 and ERK increased GADD34 gene promoter activity in myoblasts. However, ATRA induced only ERK phosphorylation in GM and C2C12 myotubes cultured with low-glucose DMEM, not in high-glucose DMEM in Figure 2. Taken together, ATRA increases the expression of GADD34 in vivo and in C2C12 myotubes via ERK-dependent manner. The description has been added to page 7, line 14 – page 8, line 2; page 8, lines 5–12; page 10, lines 7–13.

Reviewer #1:

Comments

2. Figure 1F. While the magnitude of ATF4 knockdown is stated in the results section on page 7, the validation should be included in the figure. In addition, treatment of the siRNA for ATF4 cells should also be treated with vehicle and ATRA for completion.

Response: As requested, we added the data in Fig 1H.

Reviewer #1:

Comments

3. Figure 2A. In this panel, C2C12 in the absence of forced expression of the RARs/RXR does not stimulate transcription from the minimal Gadd34 promoter. C2C12 do express RARs and RXR and their myogenic differentiation can be stimulated by ATRA. If the effect is limited to C2C12 cells that express ectopic receptors, the importance of the finding is unclear, especially given that the baseline transcription increases with RAR addition. Validation in primary cells is required, and a more complete evaluation of the regulatory regions controlling GADD34 should be performed.

Response: There are cases in which ATRA does not affect the reporter plasmid without RAR overexpression (ex; Masuda et al. Biochem J. 2020). We think the amount of endogenous RARs/RXR in cells is not enough affect on the exogenous promoter plasmid. We have the results that RAR overexpression without ATRA treatment increases the luciferase activity of pGADD34-0.05k, which reporter plasmid does not have the sensitivity to ATRA, and discovered the response element of RAR on the GADD34 promoter in Figure S1. The description has been added to page 9, lines 6–14.

Reviewer #1:

Comments

4. (Continuation of the question about Figure 2.) While it is clear that the authors wish to advance a model that the RARs inhibit Six1 levels and therefore derepress GADD34 expression, which may be an important regulatory mechanism, the literature suggests that PPAR γ can stimulate GADD34 expression directly when a promoter construct of approximately 2kb in length is used (reference: PMID: 12881480). The authors should justify the minimal construct studied and why larger constructs were excluded.

Response: We do not have the longer promoter constructs than approximately 0.5kb in length and we could not success the cloning of over 2kb *GADD34* gene

promoter. However, considering our discovery the Six1-responsive element (MEF3-binding site) via ATRA exists from -155 to -147 on the human *GADD34* gene promoter, we think the necessity of the requested experiment is less to suggest our research story.

Reviewer #1:

Comments

5. (Continuation of the question about Figure 2.) The authors should also examine published data sets for RA treatment that may support or refute the direct regulation of the *GADD34* regulatory region in myoblasts. ChIP assay, especially in primary myoblasts, would support the model that RA does or does not interact with the regulatory regions.

Response: Unfortunately, we could not find such published data. Therefore, our research can suggest the important and novel role of ATRA in muscle.

Reviewer #1:

Comments

6. (Continuation of the question about Figure 2.) ChIP assay, especially in primary myoblasts, would support the model that RA does or does not interact with the regulatory regions.

Response: We determined that ATRA-dependent increase of *GADD34* gene promoter activity is regulated in myofiber mainly in vivo in Fig 1C and D. For this reason, we tried to test ChIP assay by collecting primary myofibers, not primary myoblasts, from GM in ATRA-treated mice. But unfortunately, we could not get enough amount of DNA to detect ChIP assay correctly. However, Our ChIP assay by ATRA-treated and Six1-overexpressed C2C12s in Fig S2C and D can support our research suggestion. The description has been added to page 13, line 14 – page 14, line 1.

Reviewer #1:

Comments

7. Figure 3. The authors propose that a single MEF3 element is responsible for transducing the effects of RA on the GADD34 promoter construct. This is supported by mutational analysis, but the mutation for site MEF3-2 overlaps both the MEF3-1 and the MEF3-3 sites. Confirmation that the mutations prevent binding of the individual sites without impacting neighbouring sites is not provided but is necessary. Is MEF3-1 and MEF3-3 binding affected by the mutations in the MEF3-2 oligo? For panel 3H, the controls must be included if this data is to be interpreted.

Response: To clarify the effects of other MEF3 sequences, MEF3-1 and MEF3-3, on MEF3-2 binding with Six1, we made three oligos that contains specific MEF3 sequences respectively in Fig 3H. Considering that MEF3-1 and MEF3-3 specific oligos (MA and MC) could not compete with the complexes, we suggest that Six1 certainly binds to MEF3-2 specific sequence. We also added the control of GADD34 Mut-M2 probe in the same EMSA experiment in Fig 3I. The description has been added to page 12, lines 1–7.

Reviewer #1:

Comments

8. Figure 4. Panel 4F should include both ATRA treatment and ectopic Six1 to fully support the mechanism proposed.

Response: As requested, we added the data in Fig S2B.

Reviewer #1:

Comments

9. (Continuation of the question about Figure 2.) ChIP should be completed to support the idea that Six1 occupancy of the Mef3-2 site is affected by ATRA treatment.

Response: We tested ChIP assay in Fig S2C and D. Notably, we could use only mouse origin C2C12 cells to test ChIP, although we use human *GADD34* gene promoter for the luciferase reporter assay in this research. Therefore, we tested 4 primer pairs which have a rich MEF3-2 similar sequences to determine the Six1-binding regions on the mouse *GADD34* promoter by ChIP assay. As a results, we got two regions which can be bound with Six1. The description has been added to page 13, line13 – page 14, line 1.

Reviewer #1:

Comments

10. Figure 5. In panel A, it is unclear why the complete data including error is not provided.

Response: As requested, we added complete data including error in Fig 5A.

Reviewer #1:

Comments

11. (Continuation of the question about Figure 2.) In panel E, samples that receive Six1 should also be treated with RA. Further, siRNA against Six1 could provide useful information instead of relying uniquely on overexpression systems.

Response: As requested, we added the data in Fig 5E.

Reviewer #1:

Comments

12. Figure 6. The labelling and legend for panel A makes it impossible to distinguish from Figure 1E, though the text suggest that this blot is from samples treated for only 3 hours. The data presented in panels C-F are problematic. The pGL3-basic vector does not contain a promoter and thus is not expected to

result in the production of luciferase. However, the authors use this vector to assess the importance of the GADD34 3'-UTR on mRNA stability in this context. There is concern that the data generated is captured from signal that is not in a range that allows for accurate interpretation. This should be repeated using a vector containing a minimal promoter that is demonstrated not to be affected by ATRA treatment.

Response: We added the statement of “3 h” above the result of western blotting in Fig 6A to distinguish from Fig 1G. Though the pGL3-basic vector does not contain a promoter as you mentioned, the basal RLU is given by background effects of pGL3-basic, which is enough to determine the effects of luciferase reporter assay. Naturally, we cannot detect the RLU with the plasmid that does not contain the luciferase sequence. Some reports use pGL3-basic vector without promoter to determine the 3'UTR activity (Ge et al. BMC Cancer. 2019; Li et al. Aging (Albany NY). 2020). Taken together, we believe to no need to insert the promoter to pGL3-basic in this case and can suggest our hypothesis from Figure 6.

Reviewer #1:

Comments

13. Figure 7. This figure explores the biological importance of the described mechanism by forcing expression of GADD34 in C2C2 myoblasts. The results indicate that there is no difference in the ability of these myoblasts to differentiate, though there are several important problems with the data as presented. Since the authors demonstrate that the upregulation of GADD34 occurs in proliferating myoblasts and not myotubes, the authors should characterize the function of these cells more completely. Is proliferation affected +/-RA and +/-GADD34? Loss of function experiments would also be very useful. Does knockdown of Gadd34 have an impact on myogenic differentiation? In addition, Six1 is known to be expressed in myoblasts and its expression decreases with differentiation

(<https://academic.oup.com/nar/article/38/20/6857/1309345>). If the model is correct, do Gadd34 levels surge in myotubes? A time course of Gadd34 expression would be beneficial for understanding. The assessment of myogenic differentiation is normally achieved by performing myosin heavy chain IIF and calculating the percentage of nuclei in MyHC+ cells as compared to total nuclei as well as myotube fusion (number of nuclei per myotube). The assessment of myotube thickness is an unreliable measure of differentiation.

Response: We added the data of time course of GADD34 expression during the C2C12 differentiation in Fig S4A. As expected, the GADD34 mRNA expression increased by differentiation. As mentioned in the introduction, ATRA controls muscle differentiation via regulation of MyoD and myogenin. Moreover, previous report showed ATRA decreased the number of nuclei and increased the percent of MYHC positive nuclei in C2C12 myotubes (Kruger and Hoffmann, 2010). It suggested that ATRA down-regulates the proliferation but up-regulates the differentiation in C2C12. To determine the effect of GADD34 overexpression on the proliferation and differentiation of C2C12 in detail, we added the data of fusion index as you suggested in Fig 7D. As a result, GADD34 overexpression did not change the number of nuclei and fusion index. Considering thickness of myotubes and major differentiation markers have no significant changes by overexpression of GADD34, we concluded GADD34 does not affect on cell proliferation and muscle differentiation without loss of function experiments.

Reviewer #1:

Comments

14. Figure 7B. Further, while there appears to be efficient differentiation based on the images in panel B, the trajectory of Pax7 expression is not as expected for controls and the authors should comment on this.

Response: We changed the primers to detect the mRNA expression of Pax7 and added the data in Fig S4C.

Reviewer #1:

Comments

15. (Continuation of the question about Figure 2.) Most importantly, the successful overexpression of GADD34 is not shown in the main figure, and without this important control, it is impossible to assess this data. In panel E, the expression for GADD34 is missing, but is a critical control for assessing the data presented. Figure 7F. The legend is missing from this panel and the mRNA levels of Gadd34 are missing but required.

Response: We added the data of successful overexpression of human GADD34 in Fig 7E and S4B. The legend is revised as you suggested.

Reviewer #1:

Comments

16. The Abstract needs to be re-written for clarity. In its present form, while the main ideas are stated, the sentence structure prevents understanding. In particular lines 15-17 are problematic.

Response: Thank you so much for useful advice. We revised the manuscripts.

Reviewer #1:

Comments

17. Additional information about the known mechanisms regulating Gadd34 expression should be added to the Introduction.

Response: Thank you for your suggestion. We added the information in page 5, lines 1-5.

Reviewer #1:

Comments

18. The area of research is important and has the potential to contribute to the mechanistic understanding of retinoic acid receptor function during myoblast differentiation. Interesting data is provided, however, the story feels at times incomplete and requiring further information to significantly advance the field. The data presented herein is derived primarily from one cell line that has been shown to not be an entirely reliable model of myogenic differentiation. This coupled with an over-reliance on overexpression systems, lack of controls for some experiments, and concerning experimental design in Figure 6, prevents significant advancement of knowledge. The authors fail to place their work in the context of the literature, including the discovery of RXR binding to the GADD34 promoter and the expression of Six1, including excellent datasets that could support binding to the GADD34 promoter region in myoblasts. A more complete alignment with the models proposed and the literature would improve the manuscript overall, though more use of primary cells and in vivo models would be required to rise this contribution to the level expected by the journal. In general the work presented is geared towards a specialty audience that is interested in mechanism of retinoic acid action in myoblasts. As a reviewer, my research aims to understand the transcriptional mechanisms regulating myogenic differentiation. I am an expert in nuclear receptor biology and retinoic acid signaling, and can appreciate the importance of the mechanistic detail proposed. I also, however, recognize that the work does not represent the cutting edge of the field in terms of techniques used, approaches, or use of available datasets.

Response: Thank you so much for your useful comments. As requested, we have tested IHC in vivo (Fig 1C), myofiber isolation (Fig 1D), ChIP assay (Fig S2C and D), detailed luciferase assay (Fig S1), and some siRNA experiments (Fig 1H, 4H and 5E). We believe these experiments progressed our research dramatically. Moreover, we discovered the difference of ATRA sensitivity in C2C12 myotubes cultured with high- or low-glucose DMEM, which is regulated via ERK dependent manner (Fig 1F, 2D–H). Although ATRA has an important

role in the transcriptional regulation as a ligand of RAR in nucleus classically, recent studies say ATRA have a non-genomic action via MAPK signaling. Our research can be the new right in the research field of ATRA and muscle.

Reviewer #2:

Comments

1. Gadd34 gene was shown to be induced in the mouse gastrocnemius muscles (mainly composed of differentiated myofibers) (Figure 1A). However, in cultured C2C12 myoblasts, Gadd34 gene was shown to be only induced in undifferentiated myoblasts but not differentiated myotubes (Figure 1D). How do you explain this obvious discrepancy?

Response: Thank you so much for your valuable suggestion. We checked ATRA-induced GADD34 localizes in myofibers mainly in gastrocnemius muscle (GM) by analyzing IHC in Fig 1C. We also revealed that ATRA increased GADD34 mRNA expression in isolated primary myofiber in Fig 1D. Therefore, we hypothesize that ATRA increases the expression of GADD34 in not only myoblasts but also myotubes. Recent reports show that there are cases in which low-glucose DMEM is better to get a similar signal with in vivo study than high-glucose DMEM, which is the most common medium to culture C2C12 classically. We confirmed that ATRA increased GADD34 mRNA expression in low-glucose DMEM cultured C2C12 myotubes in Fig 1F. To investigate the reason why the sensitivity of ATRA to the *GADD34* gene promoter activity is different in the status of C2C12 cells, we checked the effects of ATRA on MAPK signal in GM and C2C12 myotubes cultured with high- or low-glucose. As a results, ATRA-induced phosphorylation of p38 and ERK increased GADD34 gene promoter activity in myoblasts. However, ATRA induced only ERK phosphorylation in GM and C2C12 myotubes cultured with low-glucose DMEM, not in high-glucose DMEM in Figure 2. Taken together, ATRA increases the expression of GADD34 in vivo and in C2C12 myotubes via ERK-dependent

manner. The description has been added to page 7, line 14 – page 8, line 2; page 8, lines 5–12; page 10, lines 7–13.

Reviewer #2:

Comments

2. Although mouse C2C12 cells were extensively used for mechanistic studies in this manuscript, it is counter-intuitive that authors chose to use human *Gadd34* promoter for detailed transcriptional analysis in this manuscript. It is never mentioned whether some of the key cis-elements identified in this manuscript (e.g., the MEF3 site, or the ARE element in 3'UTR) are conserved in the mouse *Gadd34* promoter or not.

Response: The consensus sequence of MEF3 is determined with both ends of nucleotides strongly, but the inside 7 nucleotides have flexibility compared with both ends (Santolini et al. *Nucleic Acids Res.* 2016). Although we cannot find the completely same MEF3 sequence in the mouse *GADD34* promoter, some similar MEF3 sequences are found. Actually, we checked the binding of Six1 to MEF3 on mouse *GADD34* promoter by ChIP assay in Fig S2C and D. Although we could not determine the same MEF3 region binding with Six1 as the human *GADD34* promoter, we confirmed more than two regions on the mouse promoter. On the other hand, the ARE elements in *GADD34* 3'UTR are highly conserved in the mouse.

Reviewer #2:

Comments

3. Although overexpression of *Gadd34* in C2C12 myoblasts slightly reduced the expression of two fast-type myosin heavy chains, it is premature to claim that elevated *Gadd34* promotes muscle fiber type switch. More rigorous assays (including more in vivo assays) are required in order to draw such a conclusion.

Response: We additionally confirmed that ATRA treatment decreased the protein expressions of total MYHC and MYHC2, but not MYHC1, in the GM of mice. Considering GADD34 overexpression decreased the mRNA expression of *MYH2* and *MYH1*, but not *MYH7* in C2C12 cells in Fig 7F, ATRA-induced GADD34 protein may contribute to the acceleration of decreasing type 2 MYHC expressions strongly. The description has been added to page 18, lines 11–13; page 19, lines 13–15; page 23, lines 10–12

Reviewer #2:

Comments

4. Figure 3G: the exogenously transfected Six1 was used in EMSA to prove that the MEF3-2 site of the Gadd34 promoter was bound by Six1. As Gadd34 gene was normally repressed in C2C12 myoblasts in the absence of ATRA, one would expect that the endogenous Six1 also binds to the Gadd34-MEF3 probe. It would be more convincing to use the endogenous Six1 in such EMSA experiments.

Response: We added the EMSA data detected by siSix1 transfected C2C12 cells in Fig 4H.

Reviewer #2:

Comments

5. Figure 7D: In cultured C2C12 myoblasts, the endogenous Pax7 mRNA is known to be expressed at very low levels (Seale et al., Cell, 2000). Moreover, Pax7 gene expression further goes down upon differentiation (Zammit, et al., J Cell Sci, 2006). Therefore, it is counter-intuitive to see a gradual increase in Pax7 mRNA upon myoblast differentiation.

Response: We changed the primers to detect the mRNA expression of *Pax7* and added the data in Fig S4C.

Reviewer #2:

Comments

6. Overall, the cell culture-based assays in this manuscript were generally solid and well controlled. The effect of ATRA on Gadd34 gene induction was also clear. However, without solid *in vivo* data and a clear link of the central research question with a well-defined developmental process or disease setting, the impact and the significance of such studies are not high.

Response: Thank you so much for your helpful comments. Considering our additional experiments of IHC *in vivo* (Fig 1C), myofiber isolation (Fig 1D), and *in vitro* assays cultured with low-glucose DMEM, this research can be one of the important studies in the research field of ATRA and muscle (Fig 1F, 2D–H). As mentioned in the discussions, the plasma levels of vitamin A increase with aging and CKD. Aging and CKD has the specific muscle fiber type changes similar as shown in present study. Taken together, our discovery has a big impact on the research field of some diseases.

March 7, 2022

RE: Life Science Alliance Manuscript #LSA-2021-01345R

Dr. Masashi Masuda
Tokushima University Graduate School
Clinical Nutrition and Food Management
3-18-15 Kuramoto-cho
Tokushima 770-8503
Japan

Dear Dr. Masuda,

Thank you for submitting your revised manuscript entitled "All-trans retinoic acid changes muscle fiber type via increasing GADD34 dependent on MAPK signal". We would be happy to publish your paper in Life Science Alliance pending final revisions necessary to meet our formatting guidelines.

- Please upload all figure files as individual ones, including the supplementary figure files; all figure legends should only appear in the main manuscript file
- please add the Twitter handle of your host institute/organization as well as your own or/and one of the authors in our system
- please upload your Tables in editable .doc or excel format
- please add your main, supplementary figure, and table legends to the main manuscript text after the References section
- please note that titles in the system and manuscript file must match

FIGURE CHECKS:

- the minimum figure resolution is 300dpi

A. FINAL FILES:

B. MANUSCRIPT ORGANIZATION AND FORMATTING:

Sincerely,

March 11, 2022

RE: Life Science Alliance Manuscript #LSA-2021-01345RR

Dr. Masashi Masuda
Tokushima University Graduate School
Clinical nutrition and Food Management
3-18-15 Kuramoto-cho
Tokushima 770-8503
Japan

Dear Dr. Masuda,

Thank you for submitting your Research Article entitled "All-trans retinoic acid changes muscle fiber type via increasing GADD34 dependent on MAPK signal". It is a pleasure to let you know that your manuscript is now accepted for publication in Life Science Alliance. Congratulations on this interesting work.

DISTRIBUTION OF MATERIALS:

Again, congratulations on a very nice paper. I hope you found the review process to be constructive and are pleased with how the manuscript was handled editorially. We look forward to future exciting submissions from your lab.

Sincerely,
